# Metabonomics Analysis Reveals the Influence Mechanism of Three Potassium Levels on the Growth, Metabolism and Accumulation of Medicinal Components of *Bupleurum scorzonerifolium* Willd. (Apiaceae)

**DOI:** 10.3390/biology14050452

**Published:** 2025-04-22

**Authors:** Jialin Sun, Jianhao Wu, Alyaa Nasr, Zhonghua Tang, Weili Liu, Xiubo Liu, Wei Ma

**Affiliations:** 1School of Pharmacy, Heilongjiang University of Traditional Chinese Medicine, Harbin 150040, China; klp15sjl@nefu.edu.cn (J.S.); 17515281505@163.com (J.W.); liuweili410@126.com (W.L.); 2Biological Science and Technology Department, Heilongjiang Vocational College for Nationalities, Harbin 150066, China; 3Botany and Microbiology Department, Faculty of Science, Menoufia University, Shebin Elkoom 32511, Egypt; alyaa.abd@science.menofia.edu.eg; 4College of Chemistry, Chemical Engineering and Resource Utilization, Northeast Forestry University, Harbin 150040, China; tangzh@nefu.edu.cn

**Keywords:** potassium application, *Bupleurum*, mineral elements, metabolite profiles, growth regulation

## Abstract

Wild *Bupleurum scorzonerifolium* Willd. is over-mined, and its cultivation has issues like improper fertilization, leading to low yield and quality. This study aimed to find out how different levels of potassium fertilizer affect *Bupleurum scorzonerifolium* Willd. The researchers applied three levels of potassium fertilizer and analyzed the plant’s metabolites. They found that low and high potassium levels promoted the growth of main shoots and roots but reduced dry matter in lateral shoots and flowers. Low potassium increased saikosaponin a content, while high potassium inhibited the accumulation of some saikosaponins. The study identified key metabolites and related metabolic pathways. These findings can help farmers use potassium fertilizer more rationally, improve *Bupleurum scorzonerifolium* Willd. yield and quality, and are valuable for sustainable herbal medicine production.

## 1. Introduction

*Bupleurum scorzonerifolium* Willd., a plant of the Umbelliferae family, is an important medicinal plant species in China [1]. Its dried roots are used as medicine. Clinically, it is commonly used to treat a variety of conditions such as cold and fever, irregular menstruation, distending pain in the chest and hypochondrium, and organ prolapse [2,3,4]. The resources of *Bupleurum* are quite abundant and widely distributed. However, for a long time, commercial *Bupleurum* has been overly reliant on wild resources for supply. As a result, the yield of wild *Bupleurum* resources cannot even meet 20% of the market demand [5]. In Northeast China, there is a vast expanse of black soil, which is extremely fertile. At present, there are large-scale planting sites for *Bupleurum*. As a crop with a high potassium requirement, maize has been widely and continuously planted in Northeast China for a long time. In Northeast China, the long-term and continuous cultivation of plants with high potassium requirements in the soil, such as corn, soybeans, and rice, has led to a gradual decline in the content of available potassium in the soil [6,7].

Nitrogen (N), phosphorus (P), and potassium (K) are regarded as crucial and indispensable nutrients in the growth and development process of plants [8,9]. In the early stage of this study, our research group successfully completed the relevant studies on applying different levels of nitrogen and phosphorus fertilizers to *Bupleurum* [10,11]. Potassium (K), a macronutrient, can make up as much as 10% of plants’ dry mass. Also, it is crucial in controlling plant expansion, regulation and the operation of stomata [12,13,14,15], cell volume growth and cytoskeleton continuum within plant cells [16,17,18,19]. Plants coordinate the transport of K^+^ through channels and transporters in vivo to meet the demand for K^+^ [20,21,22]. Additionally, the presence of K^+^ in cytoplasm plays a vital role in controlling the synthesis of proteins and the activity of enzymes in root cells, achieved by preserving the pH and the charge of protein anions. In addition, adaptive changes in plant root and root hair also promote the absorption of potassium ions by potassium stress [23,24]. Plants usually cope with nutrient deficiency by distributing biomass to roots, so as to improve root cap and improve their capacity to assimilate nutrients present in the soil. Therefore, an appropriate amount of K^+^ is imperative to enhance the yield and quality of plants and improve plants’ capacity to resist biotic and abiotic stresses [25,26,27].

During the process of metabolism, plants produce a large number of metabolites. Exploring plant metabolomics can help us gain in-depth insights into the complex metabolic processes within plants and the diverse range of plant metabolites. These metabolites play a crucial role when plants respond to changes in their living environment. Previous studies on *Bupleurum* mainly focused on its response mechanisms under drought and salt—alkali stress conditions—and there is a lack of research on the response mechanisms of *Bupleurum* under the influence of nutrients [28,29,30,31]. The chemical constituents of *Bupleurum* are complex, including saponins, volatile oils, flavonoids and polysaccharides. Studies have shown that there are significant differences in metabolite profiles not only among different medicinal varieties of *Bupleurum*, but also in the investigations of metabolite profiles of different parts of *Bupleurum* [32]. In the field of plant science, the relationship between mineral elements and metabolites has always been a research hotspot. Early studies mainly focused on the impact of mineral elements on basic plant growth indicators. Nowadays, research has delved into the molecular level to explore their internal connections with metabolites. Studies have shown that an appropriate amount of potassium can promote the absorption and utilization of nitrogen and phosphorus by plants, improve the photosynthetic efficiency of crops, and thus affect the accumulation of metabolites such as sugars and starches [33]. In addition, medium—and trace—elements, such as calcium, magnesium, iron, and zinc, are also closely related to plant metabolites. Calcium, as the second messenger in cell signal transduction, participates in regulating the plant hormone signaling pathway and affects plant growth, development, and metabolite synthesis [34]; magnesium is the core component of chlorophyll, which is directly related to the process of photosynthesis and thus affects the production of photosynthetic products [35]; iron and zinc, as cofactors of multiple enzymes, participate in numerous redox reactions and metabolic processes in plants and have a significant impact on the types and contents of plant metabolites [36,37]. Potassium is closely related to the absorption and transport of other mineral elements, and the interaction between mineral elements and key metabolites may have an important impact on the growth, development, and quality of *Bupleurum*.

The objectives of this study are as follows: (1) To evaluate the effects of different potassium levels on the yield and quality trait parameters of the aboveground and underground parts of *Bupleurum*; (2) To reveal the impact of potassium fertilizer on the content of saikosaponins in different tissues of *Bupleurum*; (3) To explore the changes in the metabolite profiles of different tissues of *Bupleurum* caused by potassium application; (4) To screen the key metabolites and metabolic pathways involved in the regulation of biosynthesis; (5) To construct a visual model of the relationships among potassium fertilizer, metal content, saponin content, and primary metabolism. This is aimed at providing a theoretical model for the regulation of plant nutrients during the application of potassium fertilizer.

## 2. Materials and Methods

### 2.1. Design of Experiment

The experimental site is located in the *Bupleurum* Planting Research Base in Daqing, China (47°18′ N, 124°87′ E). It belongs in the continental monsoon climate in the North Temperate Zone, with a large temperature difference among the four seasons. The average annual temperature during the growing period from 2022 to 2024 is 4 °C, the average annual precipitation is 417.2 mm, and the average annual sunshine duration is 2807 h. The altitude of the experimental site ranges from 142.7 to 152.4 m. The chemical properties of the soil are as follows: PH value is 7.78, the content of organic matter is 4.86 g/kg; the total nitrogen content is 1.64 g/kg; the available nitrogen content is 0.21 mg/kg, the available phosphorus content is 27.4 mg/kg, and the available potassium content is 165.8 mg/kg.

The research object of this project is two-year-old *Bupleurum* scorzonerifolium. Potassium chloride (KCl) is used as the potassium fertilizer. Potassium chloride is highly soluble in water. At 20 degrees Celsius, its solubility is 34.7%. It has stable physical properties, is not prone to caking, and is convenient for application. Meanwhile, potassium chloride is a quick-acting fertilizer that can be directly absorbed by plants. Therefore, considering the natural environmental conditions of the experimental site, it is more prudent to select potassium chloride as the potash fertilizer. *Bupleurum* is treated by adjusting the application amount of potassium fertilizer, that is, it is treated with the control group (CK), the low-potassium group (LK), and the high-potassium group [38]. The three potassium fertilizer levels are as follows: CK (0 kg/ha⁻^1^), LK (7.3 kg/ha⁻^1^), and HK (14.6 kg/ha⁻^1^). Since the soil in this experimental plot is rich in potassium content, referring to the previous fertilization amount by farmers, 7.3 kg·ha⁻^1^ is regarded as the low potassium level and 14.6 kg/ha⁻^1^ is regarded as the high potassium level. Nine flat plots are selected, and the area of each plot is (3 m × 1.2 m). Each of the three experimental groups is treated with three repetitions. Potassium fertilizer is applied once in July 2024. In August 2024, samples of Bupleurum are collected from the experimental fields with consistent geographical conditions, uniformity, and on the same slope. August was chosen as it is within the optimal harvesting period for Bupleurum in the experimental area. At the time of collection, each Bupleurum specimen was directly segmented into four parts: roots, flowers, main shoots, and lateral shoots. Every group gained three technical duplications and the weight and length of fresh products were measured at the same time. Simultaneously, a part of fresh *Bupleurum* was dried in a blast oven at 42 °C, and the remaining samples were stored at −80 °C.

### 2.2. Determination of Saikosaponin

A total of 500 mg of every organ of the *Bupleurum* roots, flowers, main and lateral shoots was mixed into 25 mL of pure methanol (HPLC, ≥99.9%). The mixtures were ultrasonically treated and filtered at 30 °C for half an hour. The residue from the filter underwent two washes with 10 mL of methanol followed by merging and drying of the filtrates. The residue was re-dissolved in pure methanol (HPLC, ≥99.9%) in a 10 mL volumetric flask. A total of 20 μL was filtered for HPLC analysis (Hitachi, Tokyo, Japan). The chromatographic conditions were as follows, time: 0~50 min, acetonitrile: 25~90%, pure water: 75~10%; time: 50~55 min, acetonitrile: 90%, pure water: 10%. The chosen chromatographic column was a diamonsil C18 (4.6 × 250 mm, 5 μm), and the temperature was 25 °C, the flow rate was 0.8 mL∙min^−1^, and the calibration wavelength of the detector was 210 nm.

### 2.3. Elemental Analysis

A total of 0.4 g of the dried Bupleurum sample was put into a conical flask. A total of 5 mL of concentrated nitric acid was added. Then, the conical flask was placed on a hot plate. The initial temperature was set at 80 °C, and the temperature was gradually increased to 150–180 °C for heating digestion. The digested solution was placed in a water bath to remove acid until the solution was reduced from 5 mL to 1 mL, and the acid removal process was completed. The solution was dissolved in a volumetric flask at a constant volume. It was then passed through a membrane measuring 0.45 μm to achieve the desired sample mixture. The contents of Na, Ca, K, Mg, Mn, Zn, Fe and Cu were determined by ICP-OES (Optima 8000, Perkin Elmer, Waltham, MA, USA) analysis, and the instrument automatically absorbed 3 mL of sample solution on the machine for analysis.

### 2.4. GC−MS Analysis

The extraction and determination methods were implemented as indicated by Jialin et al. [11]. A total of 60 mg of different plant parts (roots, main branches, lateral branches and flowers) were weighed. An amount of 540 μL of methanol and 60 μL of internal standard were added, respectively. Therewith, 300 μL chloroform and 600 μL water were added, and the above vortex and ultrasonic steps were repeated. Afterwards, 300 μL chloroform and 600 μL water were added, with continuous vortexing and sonicating. Afterwards, centrifugation was carried out at 14,000 rpm for 10 min and evaporation was carried out to dry the supernatant. After evaporation, the sample was dissolved in 400 μL methoxyamine pyridine solution, incubated for 90 min, and the temperature was set at 37 °C. Then, 60 μL Hexane and 400 μL BSTFA were added, vortexed and derivatized. The supernatant was obtained by centrifuging it at a speed of 12,000 rpm for a duration of 5 min. Temperature program: 0~125°C, heating rate 8 °C/min; 125~210 °C, heating rate 4 °C/min; 210~270 °C, heating rate 5 °C/min; 270~305 °C, heating rate 10 °C/min; final 305 °C. The electron impact ion source was always kept at 260 °C, the voltage was constant at 70 V.

### 2.5. Multivariate Statistical Analysis

Through hierarchical cluster analysis, we normalized each compound’s content value to create a heat map illustrating their relative differences. To assess the differences in biomass and metabolite concentrations among CK, LK, HK, a one-way variance analysis was utilized. After data processing, *p* < 0.05 and fold changes surpassing 1 indicated statistical relevance, and differentially expressed metabolites were identified by volcanic map. KEGG was used for analysis, and different levels of potassium fertilizer have the most obvious impact on those metabolic pathways. GraphPad Prism 9.0 was used to create the graphs involved. The related paths for visual analysis were drawn through Cytoscape version 3.7.1.

### 2.6. Experimental Instruments and Reagents

AR2130 electronic balance (Shanghai Shenglong Electronic Technology Co., Ltd., Shanghai, China); Grinding instrument (MM400, Retsch GmbH, Haan, Germany); Milli-Q ultrapure water system (Millipore, Milford, MA, USA); RE532CS rotary evaporator (Shanghai Yalong Biochemical Instrument Factory, Shanghai, China); GL-16W benchtop centrifuge (Hunan Xiangyi Experimental Instrument Development Co., Ltd., Changsha, China); Hitachi High-performance Liquid Chromatograph L-2000 (Hitachi Ltd.). UPLC-QTOF/MS Ultra-high Performance Liquid Chromatography-Quadrupole-Time-of-Flight Mass Spectrometry (Waters, Tokyo, Japan). Standard substances of sodium (Na), potassium (K), calcium (Ca), magnesium (Mg), copper (Cu), zinc [39], iron (Fe) and manganese (Mn) (purity ≥ 98%, Beijing Wanjia Shouhua Biotechnology Co., Ltd., Beijing, China); Standard substances of saikosaponin a, saikosaponin c and saikosaponin d (purity ≥ 98%, Sichuan Mansite Biotechnology Co., Ltd., Pengzhou, China); Potassium fertilizer (Jinan Xinguan Chemical Products Co., Ltd., Jinan, China).

## 3. Results

### 3.1. Comparative Analysis of Quality and Traits of Bupleurum Under Different Potassium Levels

When potassium is deficient, the edges of *Bupleurum* leaves turn yellow, and the leaves curl irregularly, showing a withered state. The roots and stems develop poorly, and the plant’s resistance to diseases and pests weakens (Figure 1).

Figure 2A shows that applying potassium fertilizer to *Bupleurum* promotes the growth of the main shoots and roots while inhibiting the growth of lateral shoots. Compared with the CK group, the lengths of the main shoots in the LK and HK groups increased by 94.3% and 30.4%, respectively, and the lengths of the roots increased by 12.5% and 26.87%, respectively. The ratio of the plant dry weight (DR) is negatively correlated with the amount of potassium fertilizer applied, following the order of CK > LK > HK. However, the ratio of the dry weight of the underground part to that of the aboveground part of the plant (DSR) is positively correlated with the amount of potassium fertilizer applied, in the order of HK > LK > CK (Figure 2B). Applying potassium fertilizer increases the accumulation of dry matter in the roots and main shoots of *Bupleurum* while inhibiting the accumulation of dry matter in the lateral shoots and flowers (Figure 2C). Under the LK and HK conditions, the accumulation of dry matter in the roots increased by 38.70% and 18.87%, respectively. Under the LK condition, the accumulation of dry matter in the main shoots increased by 14.29%, while under the HK condition it decreased by 25.71%. Compared with the CK group, the accumulation of dry matter in the lateral shoots in the LK and HK groups decreased by 40.77% and 72.26%, respectively.

### 3.2. Total Saikosaponins Content Accumulated in Bupleurum Different Tissues Under Three Levels of Potassium Fertilization

According to the standard in the Chinese Pharmacopoeia (2020 Edition) [1], which takes the contents of saikosaponin A and saikosaponin D in the roots of *Bupleurum* as the evaluation indicators for *Bupleurum*, in order to further explore the medicinal value of *Bupleurum*, this study determined the contents of saikosaponin A and saikosaponin D in the lateral shoots, flowers, main shoots and roots of *Bupleurum*. Additionally, for the first time in this research context, the content of saikosaponin C in each of these plant parts was measured, aiming to further elucidate the full spectrum of medicinal potential harbored within different parts of *Bupleurum*. The results of the determination of the content of saikosaponins in *Bupleurum* after applying different levels of potassium fertilizers are shown in Figure 3. The overall trend of the content of saikosaponin A in various parts is LK > HK > CK. The highest content of saikosaponin A in the main shoots of the LK group is 3.116 mg/g, and HK causes the content of saikosaponin A in the roots to decrease to 0.498 mg/g. Therefore, LK promotes the accumulation of saikosaponin A, while HK inhibits the accumulation of saikosaponin A. The overall content trend of saikosaponin D is roots > main shoots > lateral shoots > flower parts. Among them, the highest content of saikosaponin D in the roots of the CK group is 2.093 mg/g. Under the treatment of LK, the content of saikosaponin D in the main shoots is higher, reaching 1.585 mg/g. LK significantly promotes the accumulation of saikosaponin D in the main shoots, while the effect of HK on saikosaponin D is not obvious. The overall trend of the content of saikosaponin C in various parts is CK > HK > LK.

### 3.3. Overview of the Metabolite Profiles in Response to Three Levels of Potassium Fertilization

In order to reveal the changes of *Bupleurum* metabolites under different potassium fertilizers, the primary metabolite profiles of *Bupleurum* growing under different potassium levels were analyzed and identified by GC–MS. A total of seventy-seven metabolites were categorized into eight distinct groups, including twenty-six organic acids, two lipids, nine polyols, seven alkyl groups, twelve sugars, four glycosides, seven amino acids and fifteen additional types (Figure 4A; Table 1). Among these metabolites, the highest proportions accounted for organic acids and derivatives, which was 33.77%, while lipids had the lowest proportion with only 2.6%. According to the accumulation mode of *Bupleurum* and the relative difference of metabolites, two main clusters were obtained (Figure 4B). The metabolites of cluster Ⅰ have lower abundances in flowers and lateral shoots at both the HK and LK levels. At the CK level, the abundances in all parts are low, while only at the HK and LK levels are the abundances in the main shoots and roots relatively high. Most of the time, the metabolites of cluster Ⅱ preferentially accumulate in flowers and roots, with lower abundances in the main shoots and lateral shoots. At the LK level, the abundances in flowers and lateral shoots are increased, while the abundance in roots is decreased. At the HK level, the abundance in secondary stems is increased. The variation in metabolite levels could indicate that metabolite accumulation varies by organization specificity.

As observed in Figure 5A, the high-potassium group, the low-potassium group, and the control group in the four parts of flowers, main shoots, lateral shoots, and roots can be completely separated. The number of differential metabolites in the two parts of flowers and lateral shoots is greater than that in the main shoots and roots. PLS-DA-R^2^X(1) explains 24.6% of the characteristics of the metabolites in the flowers of *Bupleurum* (Figure 5A), and it can be found that the low-potassium group and the high-potassium group in the flowers of *Bupleurum* are completely separated; PLS-DA-R^2^X(2) explains 18.1% of the characteristics of the metabolites in the flowers of *Bupleurum* and the control group is completely separated from the high-potassium and low-potassium groups. For the lateral shoots, PLS-DA-R^2^X(1) explains 27.2% of the characteristics of the metabolites in the lateral shoots of *Bupleurum* (Figure 5B), and the low-potassium group and the high-potassium group are completely separated (Figure 5B); PLS-DA-R^2^X(2) explains 20.1% of the characteristics of the metabolites in the lateral shoots of *Bupleurum* and the control group is completely separated from the high-potassium and low-potassium groups. Similarly, PLS-DA-R^2^X(1) can separate the metabolites of the high-potassium group from those of the control group and the low-potassium group in the main stem part (Figure 5C), but the separation between the low-potassium group and the control group is not complete. PLS-DA-R^2^X(1) explains 19% of the characteristics of the metabolites in the main shoots of *Bupleurum.* The control group and the high-potassium group are completely separated by PLS-DA-R^2^X(2), and PLS-DA-R^2^X(2) explains 16.9% of the characteristics of the metabolites in the main shoots of *Bupleurum* The low-potassium group and the high-potassium group are completely separated by PLS-DA-R^2^X(1), which explains 18.5% of the characteristics of the metabolites in the roots of *Bupleurum* the control group, the low-potassium group, and the high-potassium group are separated by PLS-DA-R^2^X(2), which explains 18.2% of the characteristics of the metabolites in the roots of *Bupleurum* (Figure 5D). In this study, the model effects of the flowers, lateral shoots, and roots are all ideal, with Q^2^ > 0.5, and the difference between R^2^Y and Q^2^ is not large, indicating that the model has good predictive ability and can be used for the screening of differential metabolites (Table 2).

### 3.4. Effect of Potassium Fertilizer Application on Mineral Elements of Bupleurum

The elements in the roots, main shoots and lateral shoots of *Bupleurum* under different potassium fertilizers were determined to compare the influence of mineral elements’ absorption and transport (Figure 6). After applying potassium fertilizer, the contents of Mg, K, Ca, Mn, and Zn all conformed to the pattern of the LK > HK > CK, and the contents in the roots and lateral shoots were relatively high, while those in the main shoots were relatively low. In terms of the contents of Na and Cu, it was the LK > CK ≥ HK, and the contents were the highest in the roots and lower in the main shoots and lateral shoots. Fe was particularly special. Its contents in the main shoots and lateral shoots hardly changed, but the content in the roots increased with the increase in potassium fertilizer. With the increase in the content of potassium fertilizer, the root distribution ratios of the four elements, namely Ca, Mn, Cu, and Fe, gradually increased, showing a significant positive correlation. The root distribution ratio of Na and K was the lowest in the low-potassium group and the highest in the control group.

The total contribution rate of the principal component PC1 in the main shoot is 75.80% and that of PC2 is 20.18% (Figure 7, Table 3). The factor load of Mg is the largest in PC1, indicating that PC1 mainly reflects the information of the Mg element. The load factor of K in PC2 is the largest, showing a significant negative correlation. Mg (−0.996) and K (−0.797) are the characteristic elements in the main shoot of *Bupleurum* according to the contribution rate of factors to the total variance and the factor load. The total contribution rate of PC1 and PC2 in the lateral shoot is 88.66% and 10.33% (Figure 4B). Except for Na, the other mineral elements in the lateral shoot accumulated in the positive range, in which the factor load of K, Mg and Cu was the largest in PC1, indicating that PC1 mainly reflected the information of the K, Mg and Cu elements. The load factor of Na in PC2 is the largest, showing a highly positive correlation. K (0.997), Mg (0.997), Cu (0.997) and Na (0.763) are the characteristic elements of the lateral shoot of *Bupleurum*, according to the contribution rate of the factors. The total contribution rate of PC1 and PC2 in the roots is 77.00% and 22.99%, respectively (Figure 7). The load factor of Zn is the largest in PC1, indicating that PC1 mainly reflects the information of Zn. The factor load of Na in PC2 is the largest, showing a significant positive correlation. Zn (0.998) and Na (0.943) are the characteristic elements in the root of *Bupleurum* according to the contribution.

### 3.5. Volcano Diagram and KEGG Enrichment Analysis of Primary Differential Metabolites of Bupleurum

The changes in metabolites were analyzed by fold change (FC) and *p*-value. The results showed that a total of 18 metabolites significantly changed in HK compared with LK (Figure 8A), of which 6 metabolites were significantly up-regulated (2F, 3R and 1MS) and were concentrated in the flowers, roots and main shoot of *Bupleurum*. A total of 12 metabolites were significantly down-regulated (6LS, 6F) and were concentrated in the lateral shoots and flowers. In the KEGG enrichment analysis, these significantly different metabolites involved five metabolic pathways, including starch and sucrose metabolism, galactose metabolism, the pentose phosphate pathway, the insulin signaling pathway, galactose metabolism and streptomycin biosynthesis (Figure 8B). Comparing HK with CK, 19 metabolites changed obviously, of which 9 (3 MS, 2 F, 2 R, 2 LS) were down-regulated and 10 (3 F, 1 MS, 2 LS, 4 R) were up-regulated. The variance in the KEGG enrichment categorization with these specific pathways was as follows: glyoxylate and dicarboxyla, galactose metabolism, streptomycin biosynthesis, citrate cycle, reductive carboxylate cycle, insulin pathway (Figure 8C,D). About 23 metabolites had changes when LK compared to CK, with 18 (8 F, 8 LS, 2 R) up-regulated and 5 down-regulated metabolites (3 MS, 1 LS, 1 MS). The metabolic pathways involved above include biosynthesis of phenylpropanoids, the insulin signaling pathway, oxidative phosphorylation, glyoxylate and dicarboxylic acid metabolism, streptomycin biosynthesis, citrate cycle, reductive carboxylate cycle, galactose metabolism, starch and sucrose metabolism (Figure 8E,F). Glycerol, D-glucose, silane and copper phthalocyanine were identified as the key metabolites in response to the application of potassium fertilizer (Figure 8G). These imperative metabolites are mapped into the insulin signaling pathway, streptomycin biosynthesis, galactose metabolism along with other metabolic pathways, which ensure the metabolic regulation of *Bupleurum*.

The differential metabolites, metal contents and saikosaponins showed corresponding changes in response to different potassium levels. Based on the Pearson correlation coefficient, potassium levels, metal contents, saponin contents and metabolites were imported into Cytoscape to construct a visual network model. All the metabolites in the network are interrelated, and the degree of correlation is indicated by generating lines between relevant nodes (Figure 9). For the LK group, the grid diagram involved 28 positively correlated substances and 36 negatively correlated substances. Among them, it was significantly positively correlated only with the metals Na-R, Ca-MS and Mg-R, but significantly negatively correlated with the metal K. It was significantly negatively correlated with all parts of saikosaponin SSc and SSd-R, and positively correlated with the rest. It was significantly positively correlated with the primary metabolites such as cellobiose, D-glucose, L-rhamnose, sedoheptulose, galactopyranoside, ribitol, Cuminyl alcohol, benzoic acid, lactic acid, silane and bisphenol A monomethyl ether. In the HK grid diagram, there were 39 positively correlated substances and 27 negatively correlated substances. Among them, it was significantly positively correlated with the metals Na-LS, K-LS, Ca-LS and Mg-MS, and the remaining eight pairs were negatively correlated. It was significantly negatively correlated with saikosaponins SSc-MS, SSa-R and SSd, and positively correlated with the rest. It was significantly negatively correlated with 11 primary metabolites and positively correlated with 15 primary metabolites. As for the CK, the grid diagram included 19 positively correlated substances and 24 negatively correlated substances. Among them, it was significantly negatively correlated with the metals K-MS, Ca-LS, K-R, Ca-R and Mg-LS, and the remaining seven were negatively correlated. It was only positively correlated with SSd-F and SSd-R, and negatively correlated with the other 10 saikosaponins. It was significantly negatively correlated with the primary metabolites L-rhamnose, disiloxane and silane, and positively correlated with D-fructose, D-glucose, D-xylose, ribitol and galacturonic acid.

## 4. Discussion

Potassium plays a crucial role in various aspects of plants, including photosynthesis, signaling molecules, protein synthesis, ionic homeostasis, osmotic regulation, and enzyme activation [30,31]. The deficiency of potassium ions (K⁺) in the soil has been a persistent problem for centuries. Plants have great difficulty in obtaining sufficient potassium from the soil during their growth and development stages, and the potassium content in agricultural soils worldwide is gradually decreasing [40,41]. The rational application of potassium fertilizer can lead to the production of high-quality plant products [42,43,44]. However, due to the relatively slower yield-increasing rate of potassium fertilizer compared with nitrogen and phosphorus fertilizer [45], it has not been fully utilized.

This study shows that low potassium (LK) and high potassium levels significantly promote the growth of the main shoots and roots of Bupleurum, making the main shoots and roots sturdier. However, in the case of potassium deficiency, *Bupleurum* exhibits poor root and main shoot development, is prone to root rot, and has slender main shoots that are likely to lodge. As a key regulator of carbon metabolism, potassium enhances the transportation of photosynthetic products to sink organs such as roots and main shoots, thereby increasing root biomass and the robustness of main shoots. That is, potassium improves the root–shoot ratio of rhizome plants by enhancing the distribution of carbohydrates [46,47,48]. Under low-potassium and high-potassium levels, the dry matter accumulation in the roots of *Bupleurum* increases, highlighting the role of potassium in promoting root development. However, the effects of potassium on the dry matter of the main shoots vary. It increases by 14.29% under low-potassium conditions and decreases by 25.71% under high-potassium conditions. It can be seen that the application of potassium fertilizer has a significant impact on the growth and development of medicinal plants, and there is a threshold. Potassium exceeding this threshold will transfer resources from aboveground tissues to the roots. After KCl is dissolved in the soil solution, it will dissociate into potassium ions (K^+^) and chloride ions (CL^−^), and the roots of plants can absorb the CL^−^ in the soil solution. An appropriate amount of chlorine has certain positive effects on plant growth. However, if plants absorb an excessive amount of CL^−^, it may also have negative impacts on them. The potential influence of CL^−^ ions is a valuable aspect for future research [49,50].

The regulation of saikosaponins by different potassium levels varies. Low potassium significantly increases the contents of saikosaponin A and saikosaponin D in various parts of the plant, while high potassium inhibits the accumulation of saikosaponin A, C, and D. This indicates a complex interaction between potassium and the biosynthesis of secondary metabolites. Saikosaponins, as the main bioactive components of *Bupleurum* in traditional Chinese medicine, are synthesized through the triterpenoid pathway, which may be affected by potassium-mediated changes in enzyme activity or precursor supply (such as acetyl-CoA, mevalonic acid). This result is consistent with studies on other medicinal plants, that is, nutrient elements usually regulate the production of secondary metabolites as an adaptive response [51,52,53].

Potassium fertilizer affects the absorption and distribution of mineral elements, showing both synergistic and antagonistic effects. Under low-potassium and high-potassium conditions, the accumulation of magnesium, potassium, calcium, manganese, and zinc in the roots and lateral branches is relatively high, reflecting an improvement in nutrient mobilization. However, with the increase in potassium content, the contents of sodium and calcium in the main shoots and iron in the lateral branches decrease, indicating competitive inhibition at the root membrane, which may be achieved through shared transporters (such as K⁺/Na⁺ antiporters). This antagonistic effect is crucial for ionic homeostasis because excessive sodium or calcium can disrupt the cell’s pH value and enzyme function. The positive correlation between potassium and magnesium/zinc in the roots indicates a synergistic absorption, which may enhance photosynthesis and antioxidant capacity.

GC–MS analysis identified 77 metabolites, among which organic acids, sugars, and polyols were the most abundant categories. KEGG enrichment analysis shows that key metabolites such as D-glucose, glycerol, and silane are related to pathways such as starch/sucrose metabolism, galactose metabolism, and the insulin signaling pathway, which are crucial for energy production, osmotic regulation, and cell wall synthesis. After the application of potassium fertilizer, the contents of organic acids, sugar alcohols, and glycosides in the flowers, lateral shoots, and roots of *Bupleurum* generally increase, which may be caused by the typical responses of soluble sugars, such as glucose, sucrose, and fructose, to potassium fertilizer. Under low-potassium conditions, the contents of most sugars in the roots and leaves increase significantly. Sucrose accumulates in the leaves first and then is transported from the leaves to the roots in response to nutrient deficiency. Therefore, the more sucrose accumulates under low-potassium conditions, the more conducive it is to root growth [54,55]. The connection between primary metabolites and the accumulation of saikosaponins is closely related to pyruvate, acetyl-CoA, and glyceraldehyde 3-phosphate. High-potassium fertilizer promotes the up-regulation of D-fructose in the roots and flowers of *Bupleurum*, leading to an increase in the raw materials glyceraldehyde 3-phosphate and pyruvate for the synthesis of saikosaponins. At the same time, the down-regulation of D-glucose reduces the consumption of sucrose and accumulates more D-fructose. In addition, the significant down-regulation of citric acid and succinic acid leads to a decrease in the consumption of acetyl-CoA, resulting in an increase in the accumulation of saikosaponins. Under low-potassium conditions, the up-regulation of D-fructose in the flowers of *Bupleurum* increases the raw materials glyceraldehyde 3-phosphate and pyruvate for the synthesis of saikosaponins, and the down-regulation of citric acid in the roots reduces the consumption of acetyl-CoA. The relationship between primary metabolites and the accumulation of saikosaponins remains to be further explored.

## 5. Conclusions

This study provides an empirical basis for balancing the application of potassium fertilizer in the cultivation of *Bupleurum* to achieve the optimization of both yield and quality. Low potassium levels promote the accumulation of the main medicinal components of *Bupleurum*, saikosaponin A and saikosaponin D, as well as moderate biomass growth. In contrast, high potassium levels enhance the biomass of the medicinal root part but inhibit the synthesis of key saikosaponins, indicating a trade-off between medicinal quality and yield. In fact, the low-potassium treatment (7.3 kg/ha^−1^) represents the optimal level, as it can not only increase the dry matter content in the roots and main shoots but also boost the content of saikosaponins. Future research could explore the molecular mechanisms underlying potassium-mediated saponin biosynthesis (such as key enzyme genes and transcription factors) and verify the application effects at the field scale under different soil conditions. This study, for the first time, elucidates the molecular mechanism by which potassium affects the synthesis of saikosaponins in *Bupleurum* through regulating the flow of primary metabolism, and establishes a four-dimensional regulation model of “potassium level-element distribution-metabolic network-medicinal components”. Integrating transcriptomics and proteomics can further clarify the molecular basis of the dual roles of potassium in primary and secondary metabolism, providing a comprehensive framework for precision fertilization in the cultivation of medicinal plants. In conclusion, this study highlights the crucial role of potassium in regulating the growth, secondary metabolism, and mineral homeostasis of *Bupleurum*. It offers practical insights into sustainable fertilization strategies to improve both the yield and medicinal quality.

## Figures and Tables

**Figure 1 biology-14-00452-f001:**
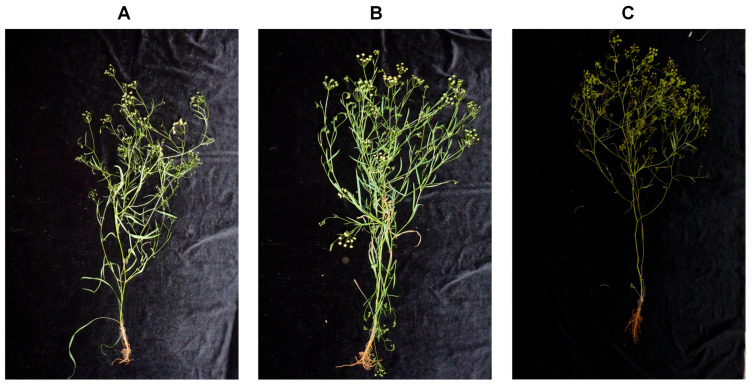
Appearance and morphology of *Bupleurum* before and after applying potassium fertilization. (**A**): control group (CK), (**B**): low-potassium group (LK), (**C**): high-potassium group [38].

**Figure 2 biology-14-00452-f002:**
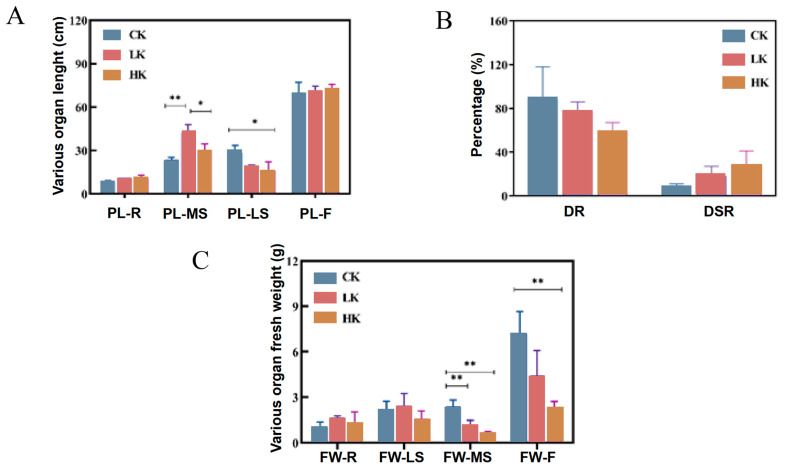
Growth performance and dry matter quality of *Bupleurum* under different levels of potassium fertilizer. (**A**) The length of each plant part of *Bupleurum*. (**B**) The drying rate and ratio of roots to shoot drying weight of *Bupleurum*. (**C**) Effect of K^+^ level on dry matter quality of *Bupleurum*. CK: control potassium group, LK: low-potassium group and HK: high-potassium group. F: flowers, MS: main shoots, R: roots, and LS: lateral shoots. PL: length of each part of the plant, FW: fresh weight of each plant parts, DR: The ratio of the weight of plants after drying to the weight before drying, DSR: Ratio of dry weight of underground part to aboveground part of plant. *p* < 0.05 and *p* < 0.01 were indicated by * and **.

**Figure 3 biology-14-00452-f003:**
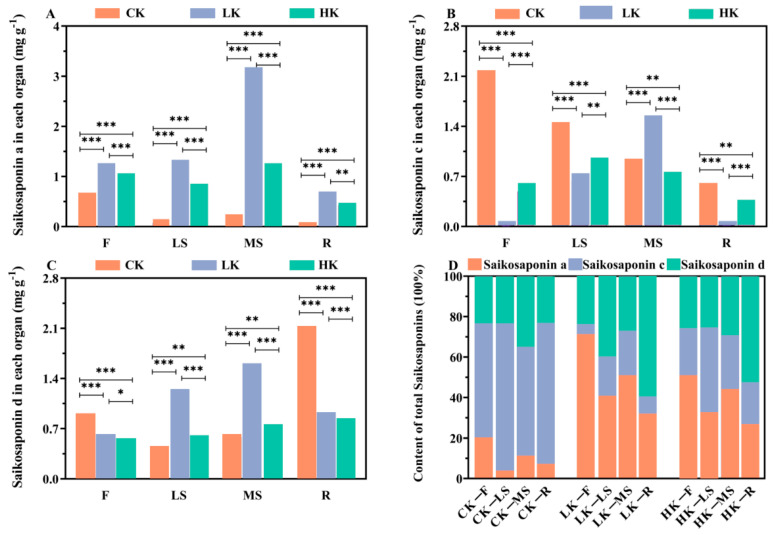
(**A**–**C**) Saikosaponin a, c, and d content under different levels of potassium fertilizer. (**D**) Percentage of saikosaponin a, c, and d in different organs of *Bupleurum* under different levels of potassium fertilizer. *p* < 0.05, *p* < 0.01 and *p* < 0.001 were indicated by *, ** and ***.

**Figure 4 biology-14-00452-f004:**
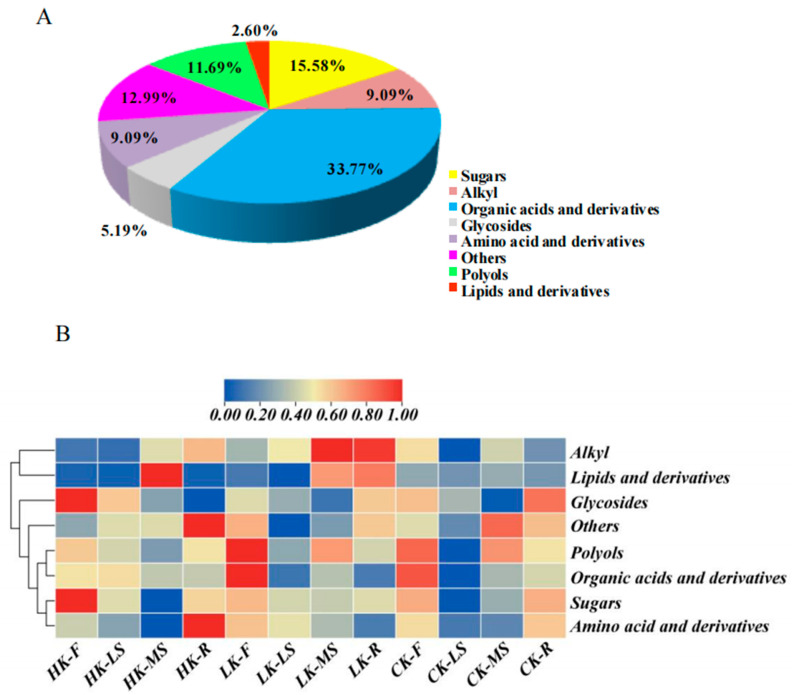
(**A**) Proportion of 77 metabolites of *Bupleurum*. (**B**) Cluster analysis thermogram of metabolites of *Bupleurum* under different K^+^ levels. CK: control potassium group, LK: low-potassium group, HK: high-potassium group. Red and blue represent down-regulated and up-regulated metabolites in turn. Beige color: abundance was 0.

**Figure 5 biology-14-00452-f005:**
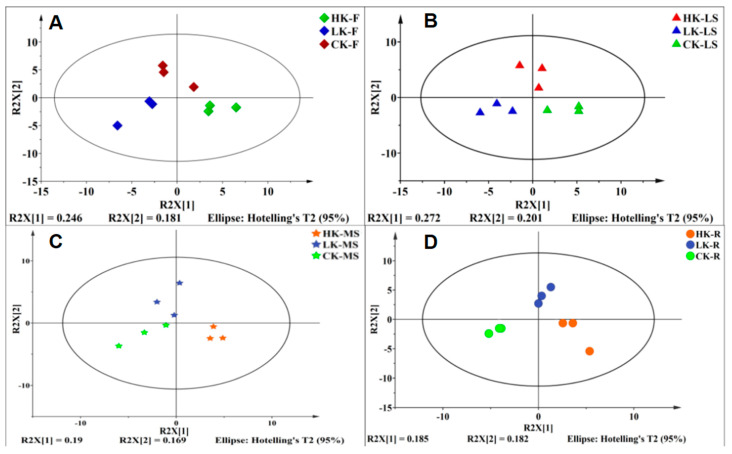
Score diagram of the metabolites of *Bupleurum* in flowers, lateral shoots, main shoots and roots. ◆, ▲, ★, ● represent F, LS, MS, R; red, blue and green represent HP group, LP group and CP group, respectively. (**A**) Score diagram of the metabolites of *Bupleurum* in root; (**B**) Score diagram of the metabolites of *Bupleurum* in main shoot; (**C**) Score diagram of the metabolites of *Bupleurum* in Lateral shoot; (**D**) Score diagram of the metabolites of *Bupleurum* in flower.

**Figure 6 biology-14-00452-f006:**
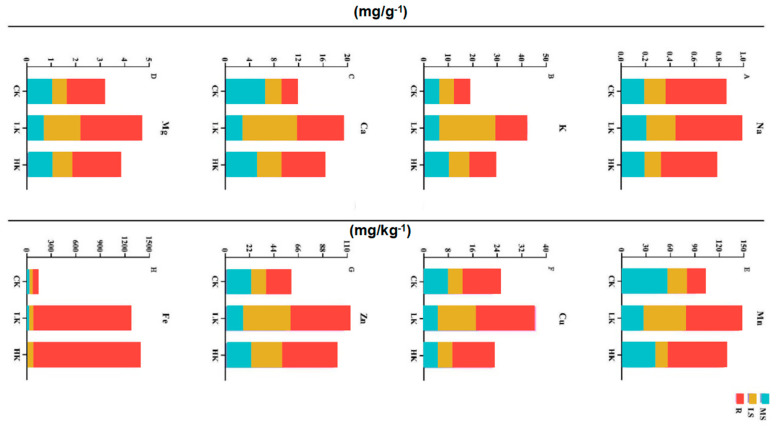
Effect of different K^+^ level on mineral elements in different organs of *Bupleurum*. (**A**) Na; (**B**) K; (**C**) Ca; (**D**) Mg; (**E**) Mn; (**F**) Cu; (**G**) Zn; (**H**) Fe. CK: control potassium group, LK: low-potassium group, HK: high-potassium group. R: roots, MS: main shoots, and LS: lateral shoots.

**Figure 7 biology-14-00452-f007:**
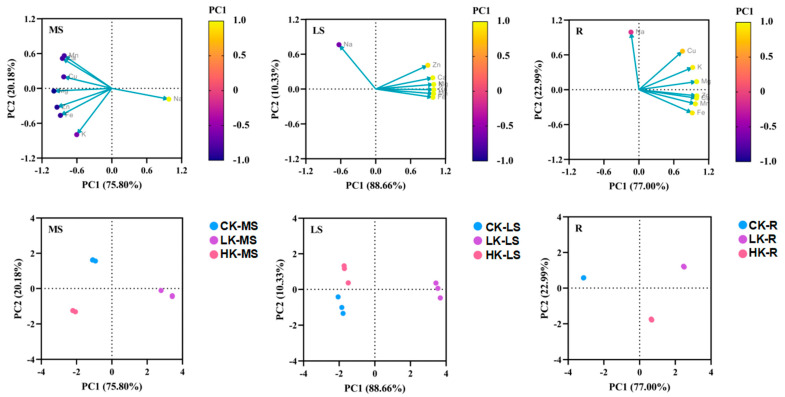
Diagram of principal component contribution rate and load matrix of mineral elements under different levels of potassium fertilizer. PC1: first principal component analysis, PC2: second principal component analysis. Yellow and purple represent higher and lower contribution rate, respectively. CK: control potassium group, LK: low-potassium group, HK: high-potassium group. R: roots, MS: main shoots, LS: lateral shoots.

**Figure 8 biology-14-00452-f008:**
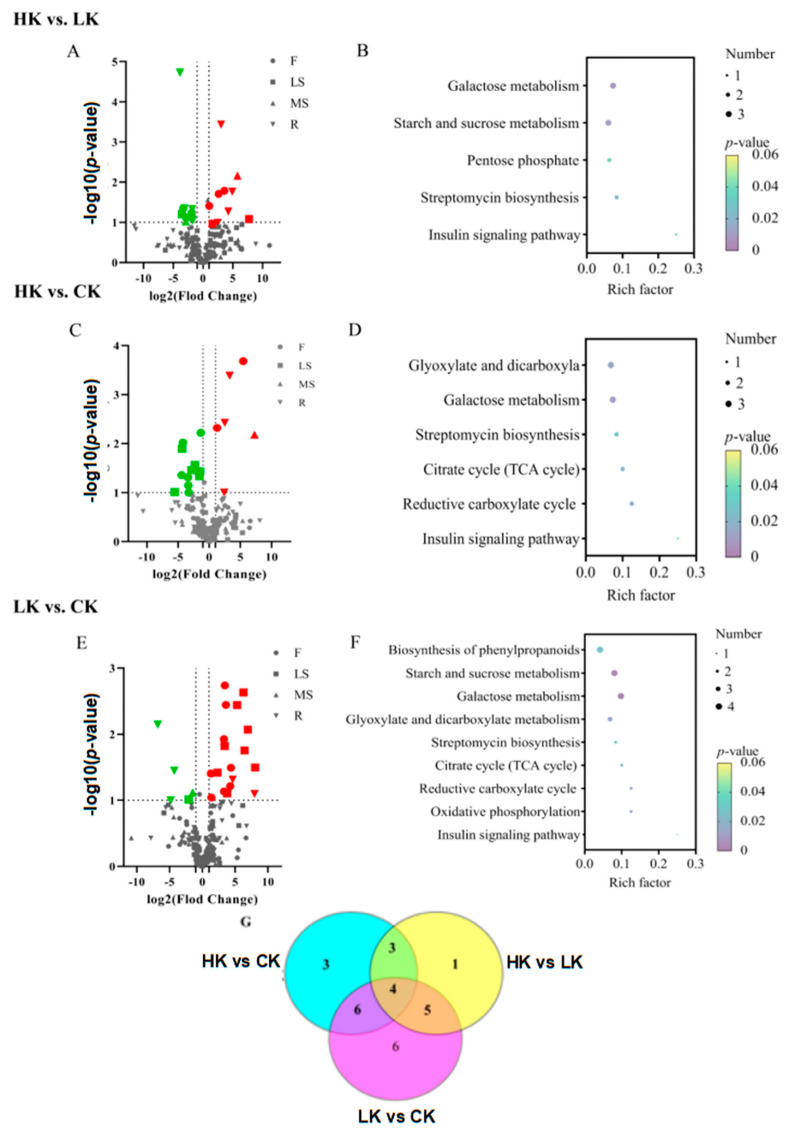
Differential metabolite volcano plots and KEGG enrichment analysis plots. Red represents a significant up-regulation of metabolites under high potassium, and green represents a significant down-regulation under high potassium; ▼, ▲, ■, ● represent R, MS, LS and F. The *p* represents the degree of enrichment. Dots represent differential metabolites, and the size is positively correlated with the quantity; (**A**,**B**) Volcano plot of differential metabolites and KEGG enrichment analysis plot for the comparison between HK and LK; (**C**,**D**) Volcano plot of differential metabolites and KEGG enrichment analysis plot for the comparison between HK and CK; (**E**,**F**) Volcano plot of differential metabolites and KEGG enrichment analysis plot for the comparison between LK and CK; (**G**) Venn diagram of metabolites.

**Figure 9 biology-14-00452-f009:**
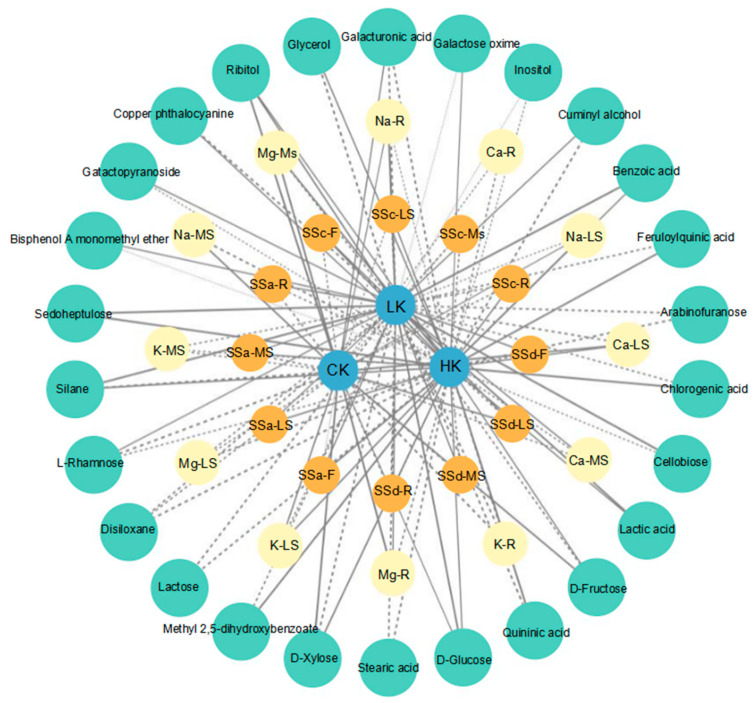
Correlation-based network analysis among potassium fertilizer, saikosaponins, metal elements and differential metabolites. All substances are represented by circles, in which the blue node represents potassium fertilizer, the green node represents primary metabolites, the yellow node represents metal elements, and the orange node represents the content of saikosaponins in different parts. Pearson coefficient is used to encode the correlation. The positive correlation and negative correlation are represented by a straight line and a dotted line. The thickness of the line means the distance of correlation.

**Table 1 biology-14-00452-t001:** Classification of 77 metabolites of *Bupleurum* after applying potassium fertilization.

Classification	Metabolites	Classification	Metabolites
Sugars (12)	α-Mannobiose	Organic acids and Derivatives (26)	Galactaric acid
	Arabinofuranose		Aminocyclopentanecarboxylic acid
	d-Cellobiose		Aminobutanoic acid
	d-Fructose		Feruloylquinic acid
	d-Glucose		Benzoic acid
	d-Xylose		Butanedioic acid
	Lactose		Caffeic acid
	Levoglucosan		Butanoic acid
	l-Rhamnose		Chlorogenic acid
	Sedoheptulose		Citric acid
	Sucrose		Gallic acid
	d-Mannose		Glyceric acid
Glycosides (4)	β-d-Galactofuranoside		Isophthalic acid
	α-d-Galactopyranoside		Lactic acid
	α-d-Glucopyranoside		Malic acid
	α-d-Lyxofuranoside		Palmitic acid
Polyols (9)	d-Mannitol		Pentanedioic acid
	Erythritol		Propanedioic acid
	Ribitol		Protocatechoic acid
	Inositol		Quininic acid
	Ethenediol		Stearic acid
	Benzenediol		Succinic acid
	Cuminyl alcohol		Sulfurous acid
	Benzylaminooctanol		2-Butenedioic acid
	Silanol		2-Propenoic acid
Amino acid and Derivatives (7)	l-5-Oxoproline		Phosphoric acid
	Valine	Lipids and Derivatives (2)	Glycerol
	Butylamine		Glycerol monostearate
	Androst-2-en-17-amine	Others (10)	Cyclohexene
	Ethanolamine		Heptabarbital
	Urea		Monopalmitin
	Copper phthalocyanine		Benzene
Alkyl (7)	Decane		Ether
	Disiloxane		Isoquinolinium
	Heptane		Phenol
	Nonane		Galactose oxime
	Silane		Methyl benzoate
	Trisiloxane		Carbamate
	Silatrane		

**Table 2 biology-14-00452-t002:** PLS-DA model parameters of *Bupleurum* GC–MS data after potassium application.

	R2X (cum)	R2Y (cum)	Q2 (cum)	R2Y-Q2
PLS-DA (F)	0.427	0.814	0.273	0.541
PLS-DA (LS)	0.598	0.934	0.550	0.384
PLS-DA (MS)	0.190	0.422	−0.006	0.481
PLS-DA (R)	0.713	0.999	0.879	0.120

Q^2^ (cum): Evaluate the prediction ability of the model through cross-validation and measure the prediction accuracy of the model for new data. The closer the value is to 1, the stronger the prediction ability of the model. Generally, Q^2^ > 0.5 indicates that the model has good prediction ability. R^2^Y-Q^2^: This value can reflect whether there is overfitting phenomenon in the model; if the value is small, the fitting effect and prediction ability of the model are more balanced.

**Table 3 biology-14-00452-t003:** Contribution rate and load matrix of mineral elements under different levels of potassium fertilizer.

MS	PC1	PC2	LS	PC1	PC2	R	PC1	PC2
Na	0.979	−0.186	Na	−0.631	0.763	Na	−0.137	0.991
K	−0.604	−0.797	K	0.997	0.080	K	0.924	0.383
Ca	−0.850	0.516	Ca	0.980	0.193	Ca	0.989	−0.145
Mg	−0.996	−0.046	Mg	0.997	0.079	Mg	0.990	0.139
Mn	−0.819	0.562	Mn	0.993	−0.081	Mn	0.970	−0.244
Cu	−0.827	0.199	Cu	0.997	−0.011	Cu	0.748	0.663
Zn	−0.943	−0.324	Zn	0.895	0.409	Zn	0.994	−0.106
Fe	−0.884	−0.466	Fe	0.983	−0.143	Fe	0.916	−0.400
Total variation explained	75.80%	20.18%		88.66%	10.33%		77.00%	22.99%

## Data Availability

The original contributions presented in the study are included in the article, further inquiries can be directed to the corresponding authors.

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
