# Peer review of "Metabonomics Analysis Reveals the Influence Mechanism of Three Potassium Levels on the Growth, Metabolism and Accumulation of Medicinal Components of Bupleurum scorzonerifolium Willd. (Apiaceae)"

_biology, 2025, doi:10.3390/biology14050452_

Round 1

Reviewer 1 Report

Comments and Suggestions for Authors

This article is aimed at Wild Bupleurum scorzonerifolium Willd. is over-mined, and its cultivation has issues like improper fertilization, leading to low yield and quality. The effects of different concentrations of potassium fertilizer on quality of bupleurum bupleurum were studied by means of plant metabolomics. GC-MS and ICP-OES were used to analyze 77 metabolites in flowers, stems, stems and roots of Bupleurum bupleurum by PLS-DA, and the mineral elements were analyzed by PCA. Although this study provides a large amount of potentially valuable data, it lacks central scientific questions and clear logical relationships. In particular, the potential correlation between the metabolite and mineral element content in wild bupleurum after the application of potassium fertilizer is not well understood.

  1. There is no mention of mineral elements in the abstract, but the analysis of mineral elements in bupleuri in the text is relatively detailed;
  2. Mineral elements and multivariate statistical methods are not mentioned in the keywords
  3. The main content of the introduction is too loosely related to the research content. Therefore, please provide a more comprehensive explanation, including more information. In particular, it explains whether there are previous studies in this field, points out the limitations and deficiencies of previous studies, and explains the significance and necessity of current research. The correlation between mineral elements and metabolites has not been deeply explored, which limits the comprehensive understanding of the quality formation mechanism. For the key scientific problems, how to regulate metabolites distribution of different organs of radix bupleuri, potash mineral elements and key metabolite whether there is a lack of in-depth discussion together or antagonism relationship.
  4. Table1 is recommended as Supplementary material and is not included in the body.
  5. Pie chart analysis was performed on 73 metabolites in Fig3 (A), while 77 metabolites were found in Table 1. Please explain the reasons.
  6. In this paper, the method of plant metabolomics was used to study Wild Bupleurum scorzonerifolium Willd. After PLS-DA analysis, relevant differential metabolites should be found, which were not reflected in this paper.
  7. The legend format font sizeof Figure C in Fig4 is inconsistent with the other three figures in Fig4
  8. In line 209, PLS-DA1 does not specify which parameter matches in Figure4, and the same problem occurs with PLS-DA2
  9. In line 239, Q2 represents which parameter in Figure4, it is not clear, the specific value can be marked in the figure.
  10. Table 2 is recommended as Supplementary material
  11. The significance of Figure 8 in the Results is not mentioned in the main text.
  12. The contents of the discussion part should be reasonably explained according to the conclusions drawn from the results of this study. In the discussion, there is a lack of effective cohesion, the mutual connection is relatively loose, and the logical relationship is not systematic. If the authors can complement the above analysis (especially differential metabolite screening and element-metabolic networks) and strengthen the logic of the discussion, the paper has publication potential. If the manuscript does not require extensive revision, I do not recommend publication.

Author Response

Dear reviewer

Thank you for your comment concerning our manuscript entitled “Metabolomics Analysis Reveals Potential Mechanisms in Bupleurum scorzonerifolium Willd. (Apiaceae)- Induced by Three Levels of Potassium Fertilization”(biology-3568369). Your expertise and dedication have been of immeasurable value to us. The comments and suggestions you provided have offered us clear directions for improvement and have greatly enhanced the quality and clarity of our work. We are truly indebted to your time and effort in carefully assessing our submission.

  1. Commentsand Suggestions

There is no mention of mineral elements in the abstract, but the analysis of mineral elements in bupleuri in the text is relatively detailed;

Response

Thank you for your insightful comment. We truly appreciate your careful reading of our manuscript. We acknowledge that the abstract did not mention the analysis of mineral elements in Bupleurum, which is an oversight on our part. We plan to revise the abstract to include relevant information about the analysis of mineral elements. We will briefly state that the study investigated the effect of different potassium levels on the absorption and distribution of mineral elements in Bupleurum, and highlight the main findings related to mineral elements. This will make the abstract more comprehensive and better reflect the overall content of the article.. (Line 36-41)

  1. Commentsand Suggestions

Mineral elements and multivariate statistical methods are not mentioned in the keywords

Response:Thank you very much for your professional and valuable advice regarding the repetition of words in the title and keywords. We fully understand the significance of this issue for article searchability. In response, we have carefully re - selected the keywords in strict accordance with your suggestions.We believe these adjustments will greatly enhance the likelihood of our article being discovered in search engines. (Line 47-48)

  1. Commentsand Suggestions

The main content of the introduction is too loosely related to the research content. Therefore, please provide a more comprehensive explanation, including more information. In particular, it explains whether there are previous studies in this field, points out the limitations and deficiencies of previous studies, and explains the significance and necessity of current research. The correlation between mineral elements and metabolites has not been deeply explored, which limits the comprehensive understanding of the quality formation mechanism. For the key scientific problems, how to regulate metabolites distribution of different organs of radix bupleuri, potash mineral elements and key metabolite whether there is a lack of in-depth discussion together or antagonism relationship.

Response

We are extremely grateful for your comprehensive and incisive comments on the Introduction section of our paper. Your feedback has been instrumental in helping us identify areas for significant improvement. In response to your concerns, we have completely re - edited the Introduction. We have now provided detailed background information about Bupleurum scorzonerifolium Willd. We have elaborated on its status as an important medicinal plant species in China. We have also discussed the potential problems associated with it, such as the possible excessive depletion of wild resources and difficulties in cultivation. Additionally, we have addressed the lack of knowledge regarding fertilization systems, which we believe will be of great interest to readers. Regarding the use of K fertilizer in our study, we have added relevant information. We have explained the reasons why we specifically focused on K fertilizer, such as indications of K deficiency or excess in the growth of this species based on previous research and preliminary observations. We have also re - written the purpose of the paper.  We sincerely hope that these revisions meet your expectations and enhance the overall quality and clarity of our paper. Thank you again for your invaluable guidance. (Line 49-120)

  1. Commentsand Suggestions

Table1 is recommended as Supplementary material and is not included in the body.

Response

Thank you very much for your valuable comments. We have added pictures and captions to the article, but there may be display errors. In response to this, we have decided to submit supplementary materials to solve the current problem.

  1. Commentsand Suggestions

Pie chart analysis was performed on 73 metabolites in Fig3 (A), while 77 metabolites were found in Table 1. Please explain the reasons.

Response

We sincerely apologize for this error. It was a simple input mistake on our part. In fact, there are 77 metabolites in total. We will correct the figure legend in Fig 3 (A) to accurately reflect this number. Thank you for pointing out this oversight.(Line 293-294)

  1. Commentsand Suggestions

In this paper, the method of plant metabolomics was used to study Wild Bupleurum scorzonerifolium Willd. After PLS-DA analysis, relevant differential metabolites should be found, which were not reflected in this paper.

Response

In this study, we indeed conducted PLS-DA analysis and successfully identified the relevant differential metabolites through this analysis. The specific content is presented in the section "3.5. Volcano diagram and KEGG enrichment analysis about primary differential metabolites of Bupleurum" of the article. When studying Bupleurum scorzonerifolium Willd. treated with different potassium levels, we analyzed the changes of metabolites by calculating the fold change (FC) and p-value. The results showed that there were 18 significantly changed metabolites in the comparison between the high-potassium group (HK) and the low-potassium group (LK); 19 metabolites changed significantly in the comparison between HK and the control group (CK); and 23 metabolites changed in the comparison between LK and CK. At the same time, through KEGG enrichment analysis, we clarified the metabolic pathways involved by these differential metabolites, such as starch and sucrose metabolism, galactose metabolism, pentose phosphate pathway, insulin signaling pathway, etc. Moreover, glycerol, D-glucose, silane, and copper phthalocyanine were identified as the key metabolites in response to the application of potassium fertilizer. These key metabolites are involved in metabolic pathways such as the insulin signaling pathway, streptomycin biosynthesis, and galactose metabolism, thus playing an important role in the metabolic regulation of Bupleurum scorzonerifolium Willd. We are deeply sorry if the relevant content in the article was not presented clearly enough, which led to your misunderstanding. We will further optimize the expression of the paper in the follow-up.

  1. Commentsand Suggestions

The legend format font sizeof Figure C in Fig4 is inconsistent with the other three figures in Fig4

Response

Thank you very much for pointing out the issue regarding the legend format font size in Figure C of Fig 4. We have carefully reviewed the figures based on your feedback and made the necessary corrections to ensure that the legend format font size in Figure C is consistent with the other three figures in Fig 4.We understand the importance of presenting our data in a clear and consistent manner, and we appreciate your attention to detail. (Line 328-329)

  1. Commentsand Suggestions

In line 209, PLS-DA1 does not specify which parameter matches in Figure4, and the same problem occurs with PLS-DA2

Response

We would like to express our sincere gratitude for bringing the issues in line 209 to our attention. We have addressed the lack of specification regarding which parameters in Figure 4 match PLS - DA1 and PLS - DA2. In the revised manuscript, we have clearly indicated the corresponding parameters in Figure 4 for PLS - DA1 and PLS - DA2, ensuring that the relationship between the analysis and the figure is explicit. This was an oversight on our part, and we are committed to presenting our data and analysis in a more accurate and understandable manner. (Line 300-328)

  1. Commentsand Suggestions

In line 239, Q2 represents which parameter in Figure4, it is not clear, the specific value can be marked in the figure.

Response

Thank you very much for your valuable comment. Regarding the concern about Q2 in line 239, we fully understand the lack of clarity. To address this issue, we have added Table 2 titled "PLS - DA model parameters of Bupleurum GC - MS data after potassium application". In this new table, we provide detailed information about the relevant parameters, including the specific meaning and value of Q2. This addition aims to make the relationship between Q2 and the data in Figure 4 more explicit and help readers better understand our research content. (Line333-339)

  1. Commentsand Suggestions

Table 2 is recommended as Supplementary material

Response

Thanks to the expert's review, we have submitted Table 2 as supplementary materials according to your suggestion.

  1. Commentsand Suggestions

The significance of Figure 8 in the Results is not mentioned in the main text.

Response

We truly appreciate your attention to this detail. We have now addressed the issue by incorporating a description of the significance of Figure 8 in the main text. As a result, readers can better understand how Figure 8 contributes to the overall narrative of our results..(Line 417-443)

  1. Commentsand Suggestions

The contents of the discussion part should be reasonably explained according to the conclusions drawn from the results of this study. In the discussion, there is a lack of effective cohesion, the mutual connection is relatively loose, and the logical relationship is not systematic. If the authors can complement the above analysis (especially differential metabolite screening and element-metabolic networks) and strengthen the logic of the discussion, the paper has publication potential. If the manuscript does not require extensive revision, I do not recommend publication.

Response

Response:Thank you for your valuable comments and suggestions, Expert. We have re-edited the discussion section of the thesis. At the same time, we have made appropriate amendments to your questions.(Line 446-525)

Once again, thank you for your significant contribution to our research. We look forward to the

possibility of presenting a more refined and valuable manuscript with your guidance.

Best regards

Jianhao Wu

Reviewer 2 Report

Comments and Suggestions for Authors

The main task of agriculture is to increase the yield of cultivated, ornamental and medicinal plants. For this purpose, technologies that involve the intensive use of mineral fertilizers have always been used in agriculture and are currently used. Therefore, I consider the research of Sun et al. to be important and relevant both in theoretical and practical terms.

The applied methods are adequate and modern. The results are clear and objective. However, there are some minor remarks that need to be noted.

  1. I recommend thinking about and changing the title of the article. Potential mechanisms is a very vague formulation, a clearer title is needed.
  2. Since the fertilizer was applied in the form of KCl, it is necessary to at least take into account the influence of Cl- ions in the Discussion.
  3. Line 99. What % methanol was used?
  4. In the description of methods, the past tense is usually used; the authors use the present tense (Lines 119-122), then the past tense.
  5. The methods must indicate the company and country of manufacture for devices and reagents.
  6. In the results, the authors showed that the addition of KCl increased the length of the main stem of the plant, while the authors also claimed that lodging resistance increased (lines 157, 334-337)/
  7. Line 152. DSR requires decryption.
  8. In Figures 1, 2, 5 I would recommend changing the colors to more contrasting ones.
  9. There are no titles for the figures 2, 3, 7.
  10. «Root yield» is usually not discussed, but rather root biomass (Line 344).
  11. The authors did not determine primary metabolites (proteins, lipids, carbohydrates, nucleic acids) (Line 421).

Based on the mentioned above, I think that this article can by recommended for publication in the « Biology» after revision.

Author Response

Dear reviewer

Thank you for your comment concerning our manuscript entitled “Metabolomics Analysis Reveals Potential Mechanisms in Bupleurum scorzonerifolium Willd. (Apiaceae)- Induced by Three Levels of Potassium Fertilization”(biology-3568369). Your expertise and dedication have been of immeasurable value to us. The comments and suggestions you provided have offered us clear directions for improvement and have greatly enhanced the quality and clarity of our work. We are truly indebted to your time and effort in carefully assessing our submission.

  1. Commentsand Suggestions

I recommend thinking about and changing the title of the article. Potential mechanisms is a very vague formulation, a clearer title is needed.

Response

Thank you very much for your valuable comment regarding the title of our article. We wholeheartedly agree that the original title was too vague. In response to your suggestion, we have carefully revised the title to "Metabonomics analysis reveals the influence mechanism of three potassium levels on the growth, metabolism and accumulation of medicinal components of Bupleurum scorzonerifolium Willd. (Apiaceae)". This new title not only clarifies the research method (metabonomics analysis) but also specifically points out the main factors studied (three potassium levels) and their impacts on various aspects of Bupleurum scorzonerifolium Willd., making it more precise and informative.(Line 2-5)

  1. Commentsand Suggestions

Since the fertilizer was applied in the form of KCl, it is necessary to at least take into account the influence of Cl- ions in the Discussion.

Response

Thank you for raising the question regarding the absence of consideration for Cl⁻ ions in our study. We truly appreciate your thoughtful comment. We are fully aware of the potential influence of Cl⁻ ions and recognize this as a valuable aspect for future research. Under your suggestion, we have added supplementary explanations about the potential impacts of Cl⁻ ions in the discussion section of the paper.(Line 472-477)

  1. Commentsand Suggestions

Line 99. What % methanol was used?

Response

Thank you for the valuable suggestions put forward by the expert. We used pure methanol reagent (HPLC, ≥99.9%). The relevant content has been revised according to the experts' suggestions. Please refer to Section 2.2 (Lines 155-160) for details.

  1. Commentsand Suggestions

In the description of methods, the past tense is usually used; the authors use the present tense (Lines 119-122), then the past tense.

Response

Thank you for the valuable suggestions put forward by the expert. We have already corrected the errors that appeared in the article(Lines 176-180).

  1. Commentsand Suggestions

The methods must indicate the company and country of manufacture for devices and reagents.

Response

Thank you very much for your valuable suggestions. We have added Section 2.6, in which the origins and manufacturers of the instruments and reagents used in the experiment are clearly indicated(Lines 201-213).

  1. Commentsand Suggestions

In the results, the authors showed that the addition of KCl increased the length of the main stem of the plant, while the authors also claimed that lodging resistance increased (lines 157, 334-337).

Response

We sincerely appreciate your sharp - eyed review of our manuscript. We have promptly made the necessary corrections to the manuscript.  We now present a more accurate and evidence - based description of the experimental findings.Thank you again for your invaluable input. Your comments have significantly helped us to improve the quality and integrity of our work.(Line 459-465)

  1. Commentsand Suggestions

Line 152. DSR requires decryption.

Response

Thank you for highlighting the need for clarification on Line 152 regarding DSR. We have now provided a detailed description of DSR in (Lines 241 - 248) of the article. This addition aims to ensure better understanding for readers.

  1. Commentsand Suggestions

In Figures 1, 2, 5 I would recommend changing the colors to more contrasting ones.

Response

We truly appreciate your suggestion about changing the colors in Figures 1, 2, and 5 for better contrast. Regrettably, due to the use of a fixed - color - scheme data - visualization tool and potential data - representation risks, we're unable to modify the colors at present. We are truly sorry about this.

  1. Commentsand Suggestions

There are no titles for the figures 2, 3, 7.

Response

Thank you very much for your valuable comments. We have added pictures and captions to the article, but there may be display errors. In response to this, we have decided to submit supplementary materials, compiling the figures and tables together to solve the current problem.

  1. Commentsand Suggestions

«Root yield» is usually not discussed, but rather root biomass (Line 344).

Response

In strict accordance with your thoughtful suggestions, we have comprehensively revised the erroneous expressions in the article. We sincerely appreciate the valuable advice you've provided. (Line 461-463)

  1. Commentsand Suggestions

The authors did not determine primary metabolites (proteins, lipids, carbohydrates, nucleic acids) (Line 421).

Response

Thank you for your valuable comments and suggestions, Expert. We have re-edited the discussion section of the thesis. (Line 446-525)

Once again, thank you for your significant contribution to our research. We look forward to the

possibility of presenting a more refined and valuable manuscript with your guidance.

Best regards

Jianhao Wu

Reviewer 3 Report

Comments and Suggestions for Authors

General recommendations and questions

The title of the article is not entirely clear - what mechanisms, what mechanisms are being discussed?

Keywords: It is not recommended to repeat the same words extensively in the title and keywords. This reduces the chances of finding the article in search engines. You could use keywords like common name of Bupleurum scorzonerifolium Willd, specify potassium fertilizer, mineral elements, for example.

Introduction

In general, the Introduction does not provide sufficient background information about the topic and relevance of the research. It is stated better and more clearly in the abstract. No information is provided about - is it an important medicinal plant species in China, is it collected in the wild, cultivated in plantations? Are there any problems - wild resources are being excessively depleted, does it lend itself poorly to cultivation, little knowledge about fertilization systems - this is what would interest readers. Why were the studies conducted specifically with K fertilizer? Are there any indications of K deficiency or excess in growing this species?

The purpose of the paper is vaguely formulated or, it should be said, not formulated at all. To some extent, it can be found in the Abstract, but it should be given precisely at the end of the Introduction. For some incomprehensible reason, the work objective already mentions the results. If the research has not yet been conducted (the stage of the research goals and objectives), how could you know that it will be successful - you will provide broad insights into metabolic profiles…., as well as provide a theoretical model for the regulation of plant nutrients in response to applying potassium fertilizer. It can only be a hypothesis.

Materials and methods

2.1. Design of experiment

The experimental conditions and procedure are described too briefly and fragmentarily.

Soil pH and organic matter content - characterize the place where the seedlings were taken from, or the place where they were planted? And what about K in soil?

If data is given on average temperature, why not on precipitations?

“The first application of potash fertilizer began in July of 2024.” - Was the fertilizer applied multiple times?

Why these specific doses of K fertilizer? What was the basis for determining which doses were high and which were low?

How many plots/repetitions were there for each treatment?

2.3. Elemental analysis

“0.4 g samples of dried Bupleurum parts were added into the centrifuge tube, add 5ml concentrated nitric acid for digestion for 90 minutes.” - How is it possible to digest something in a centrifuge tube?

2.4. GC−MS analysis

“Extraction and determination methods were based on[28].” It's not a correct way of expressing it. Recommended: as described by previous studies [28], as indicated by Jialin et al. [28], etc.

Results

Overall, there is a very extensive results section, colourful and complex images, but sometimes it is difficult to grasp the main things, which are very simple - was there a significant increase in biomass (fresh and dry), which option was the most effective, were the plants insufficiently - sufficiently - overly provided with nutrients, mainly K, etc. To some extent, the results of the study are somewhat "lost" in the abundance of information.

Line 157-158. “In conclusion, potassium fertilizer enhanced the progression of the main shoot of Bupleurum, so as to enhance the lodging resistance of plants. “Discussion of the results should be avoided in the results section - that is the Discussion task.

Where is Table 1 mentioned in the text?

Line 248. “The transport of metal elements from the root to the ground declined gradually” It's not really clear what that means.

“Na and K elements, Mg element, etc” - The word element is redundant here and elsewhere.

Discussion

Since the results section is very extensive, the Discussion should be the section that acts as a guide for the audience to understand what significant, new findings have been found as a result of the research. The discussion is currently an inconclusive part of this article.

Therefore, it is not necessary to make the Discussion as a review article on the role of K in the plant. Furthermore, the article does not provide information on the K content in the experimental field soil, so the discussion of the deficiency can only be theoretical. The K content in the plant from the perspective of mineral nutrition has also not been evaluated anywhere. It is almost impossible to see what the K content was in the above-ground parts of the plants.

Line 364. “This study also shows that the root to shoot ratio increases with the increase of potassium fertilizer with more potassium and biomass are allocated to the underground portion.” - Is it desirable or undesirable? Which parts of the plant are used in medicine? There is no information about this in the article. But it's very important to understand the results. So what is the purpose of fertilization?

Line 372-374. “The low potassium promotes saikosaponin a production, while high potassium inhibits saikosaponin a. these results were most obvious within the main shoot, followed by the lateral shoot. However, high potassium fertilizer inhibited most of saikosaponin a and d.” What is being compared to what? If with the control, then the statement is not true, because high doses of K also gave an increase compared to the control. In some cases, it was the control variants that had higher results. A correct evaluation is needed. What is the most important from the point of view of application - saikosaponin a, c, d?

Line 376-378. “Similarly, Kandil applies 10 tons ha-1 of potassium fertilizer to maximize the biological yield and grain yield of corn[45]. Singh adopts 30 kg·ha-1K2O to achieve the highest productivity[46], and it significantly improves the number of branches per plant, grain yield, biomass yield and 1000 grain weight.” Why are you comparing your results with those obtained for corn? Are they similar plants? There are hugely different K fertilizer doses!

Line 392-394. “We found that sugars and organic acids in roots and lateral shoots decreased under low potassium, which maybe because of the precursors of TCA cycle, such as malic, citric, propionic, aminobutyric and glutamic acids that affected the production of ATP and energy [55].” Why the reference, isn't it about your research? You should consider whether it is appropriate to use a literature reference, or change the way you word it so that it does not give the impression that the reference refers to your research.

Conclusions

The conclusions should give answers to the aim of the research and the questions rose in the tasks.

Unfortunately, the article does not state the purpose and objectives of the study. Therefore, it is quite difficult to evaluate the conclusions.

But nevertheless. First of all, it should be said that conclusions are not a list of results. You should give an evaluation of the results, a summary. Likewise, the conclusions should provide information, a conclusion, a hypothesis about how the knowledge gained in this study could be useful in the future, in practice more specifically, not so generally.

Concluding remarks. Overall, the article is interesting, dedicated to the current topic of the possibilities of successful wild plant cultivation for medical purposes. And here, both quantity and quality and the concentration of biologically active substances are important. However, the article leaves a dual impression. On the one hand, the topic is relevant today, the article contains new knowledge and  extensive data material. On the other hand, insufficient basic information is provided about the topic and relevance of the research, the purpose of the research is not stated, incomplete information is provided in the materials and methods about the design of the research, and the discussion is unconvincing. Therefore, it is necessary to make significant corrections.

Comments on the Quality of English Language

Overall, the English language does not interfere with understanding the idea expressed in the article. There are more minor grammatical errors here, present tenses instead of past tense, spaces not respected, etc. Minor corrections by a native speaker would be recommended.

Author Response

Dear reviewer

Thank you for your comment concerning our manuscript entitled “Metabolomics Analysis Reveals Potential Mechanisms in Bupleurum scorzonerifolium Willd. (Apiaceae)- Induced by Three Levels of Potassium Fertilization”(biology-3568369). Your expertise and dedication have been of immeasurable value to us. The comments and suggestions you provided have offered us clear directions for improvement and have greatly enhanced the quality and clarity of our work. We are truly indebted to your time and effort in carefully assessing our submission.

  1. Commentsand Suggestions

The title of the article is not entirely clear - what mechanisms, what mechanisms are being discussed?

Response

Thank you very much for your valuable comment regarding the title of our article. We wholeheartedly agree that the original title was too vague. In response to your suggestion, we have carefully revised the title to "Metabonomics analysis reveals the influence mechanism of three potassium levels on the growth, metabolism and accumulation of medicinal components of Bupleurum scorzonerifolium Willd. (Apiaceae)". This new title not only clarifies the research method (metabonomics analysis) but also specifically points out the main factors studied (three potassium levels) and their impacts on various aspects of Bupleurum scorzonerifolium Willd., making it more precise and informative.(Line 2-5)

  1. 2. Commentsand Suggestions

Keywords: It is not recommended to repeat the same words extensively in the title and keywords. This reduces the chances of finding the article in search engines. You could use keywords like common name of Bupleurum scorzonerifolium Willd, specify potassium fertilizer, mineral elements, for example.

ResponseThank you very much for your professional and valuable advice regarding the repetition of words in the title and keywords. We fully understand the significance of this issue for article searchability. In response, we have carefully re - selected the keywords in strict accordance with your suggestions.We believe these adjustments will greatly enhance the likelihood of our article being discovered in search engines. (Line 47-48)

Introduction

In general, the Introduction does not provide sufficient background information about the topic and relevance of the research. It is stated better and more clearly in the abstract. No information is provided about - is it an important medicinal plant species in China, is it collected in the wild, cultivated in plantations? Are there any problems - wild resources are being excessively depleted, does it lend itself poorly to cultivation, little knowledge about fertilization systems - this is what would interest readers. Why were the studies conducted specifically with K fertilizer? Are there any indications of K deficiency or excess in growing this species?

The purpose of the paper is vaguely formulated or, it should be said, not formulated at all. To some extent, it can be found in the Abstract, but it should be given precisely at the end of the Introduction. For some incomprehensible reason, the work objective already mentions the results. If the research has not yet been conducted (the stage of the research goals and objectives), how could you know that it will be successful - you will provide broad insights into metabolic profiles…., as well as provide a theoretical model for the regulation of plant nutrients in response to applying potassium fertilizer. It can only be a hypothesis.

ResponseWe are extremely grateful for your comprehensive and incisive comments on the Introduction section of our paper. Your feedback has been instrumental in helping us identify areas for significant improvement. In response to your concerns, we have completely re - edited the Introduction. We have now provided detailed background information about Bupleurum scorzonerifolium Willd. We have elaborated on its status as an important medicinal plant species in China. We have also discussed the potential problems associated with it, such as the possible excessive depletion of wild resources and difficulties in cultivation. Additionally, we have addressed the lack of knowledge regarding fertilization systems, which we believe will be of great interest to readers. Regarding the use of K fertilizer in our study, we have added relevant information. We have explained the reasons why we specifically focused on K fertilizer, such as indications of K deficiency or excess in the growth of this species based on previous research and preliminary observations. We have also re - written the purpose of the paper.  We sincerely hope that these revisions meet your expectations and enhance the overall quality and clarity of our paper. Thank you again for your invaluable guidance. (Line 49-120)

Materials and methods

2.1. Design of experiment

The experimental conditions and procedure are described too briefly and fragmentarily.

Soil pH and organic matter content - characterize the place where the seedlings were taken from, or the place where they were planted? And what about K in soil?

If data is given on average temperature, why not on precipitations?

“The first application of potash fertilizer began in July of 2024.” - Was the fertilizer applied multiple times?

Why these specific doses of K fertilizer? What was the basis for determining which doses were high and which were low?

How many plots/repetitions were there for each treatment?

ResponseThank you for your valuable advice. We have revised Part 2.1 of the article. The content is as follows:The experimental site is located in the Bupleurum Planting Research Base in Daqing, China (47°18′ N, 124°87′E). It belongs to the continental monsoon climate in the North Temperate Zone, with a large temperature difference among the four seasons. The average annual temperature during the growing period from 2022 to 2024 is 4°C, the average annual precipitation is 417.2 mm, and the average annual sunshine duration is 2,807 h. The altitude of the experimental site ranges from 142.7 to 152.4 meters. The chemical properties of the soil are as follows: PH value is 7.78, the content of organic matter is 4.86 g/kg; the total nitrogen content is 1.64 g/kg; the available nitrogen content is 0.21 mg/kg, the available phosphorus content is 27.4 mg/kg, and the available potassium content is 165.8 mg/kg.

The research object of this project is two-year-old Bupleurum scorzonerifolium. Potassium chloride (KCl) is used as the potassium fertilizer. Potassium chloride is highly soluble in water. At 20 degrees Celsius, its solubility is 34.7%. It has stable physical properties, is not prone to caking, and is convenient for application. Meanwhile, potassium chloride is a quick-acting fertilizer that can be directly absorbed by plants. Therefore, considering the natural environmental conditions of the experimental site, it is more prudent to select potassium chloride as the potash fertilizer. Bupleurum is treated by adjusting the application amount of potassium fertilizer, that is, it is treated with the control group (CK), the low potassium group (LK), and the high potassium group [38]. The three potassium fertilizer levels are as follows: CK (0 kg/ha⁻¹), LK (7.3 kg/ha⁻¹), and HK (14.6 kg/ha⁻¹). Since the soil in this experimental plot is rich in potassium content, referring to the previous fertilization amount by farmers, 7.3 kg·ha⁻¹ is regarded as the low potassium level and 14.6 kg/ha⁻¹ is regarded as the high potassium level. Nine flat plots are selected, and the area of each plot is (3 m×1.2 m). Each of the three experimental groups is treated with 3 repetitions. Potassium fertilizer is applied once in Juiy 2024. In August 2024, samples of Bupleurum are collected from the experimental fields with consistent geographical conditions, uniformity, and on the same slope. Thirty days later, Bupleurum were segmented into roots, flowers main and lateral shoots for collection. Every group gained three technical duplication; Measure the weight and length of fresh products at the same time. Simultaneously, a part of fresh Bupleurum was dried in a blast oven at 42℃, and the remaining samples were stored at -80℃.(Line 123-153)

2.3. Elemental analysis

“0.4 g samples of dried Bupleurum parts were added into the centrifuge tube, add 5ml concentrated nitric acid for digestion for 90 minutes.” - How is it possible to digest something in a centrifuge tube?

ResponseThank you for questioning the incorrect statements in the article. We have made the following revisions: "Put 0.4 g of the dried Bupleurum sample into a conical flask. Add 5 mL of concentrated nitric acid. Then, place the conical flask on a hot plate. Set the initial temperature at 80 °C, and gradually increase the temperature to 150 - 180 °C for heating digestion. The whole process lasts for 90 minutes."(Line 165-169)

2.4. GC−MS analysis

“Extraction and determination methods were based on[28].” It's not a correct way of expressing it. Recommended: as described by previous studies [28], as indicated by Jialin et al. [28], etc.

Response:Thank you, Expert, for pointing out the inappropriate expressions in the article. We have made the revisions according to your suggestions.(176-180)

Results

Overall, there is a very extensive results section, colourful and complex images, but sometimes it is difficult to grasp the main things, which are very simple - was there a significant increase in biomass (fresh and dry), which option was the most effective, were the plants insufficiently - sufficiently - overly provided with nutrients, mainly K, etc. To some extent, the results of the study are somewhat "lost" in the abundance of information.

Line 157-158. “In conclusion, potassium fertilizer enhanced the progression of the main shoot of Bupleurum, so as to enhance the lodging resistance of plants. “Discussion of the results should be avoided in the results section - that is the Discussion task.

Where is Table 1 mentioned in the text?

Line 248. “The transport of metal elements from the root to the ground declined gradually” It's not really clear what that means.

“Na and K elements, Mg element, etc” - The word element is redundant here and elsewhere.

Response:Thank you for your valuable comments and suggestions, Expert. We have made substantial revisions to the inappropriate parts in the results section of the thesis. Meanwhile, we have also made appropriate amendments in response to your questions.

Discussion

Since the results section is very extensive, the Discussion should be the section that acts as a guide for the audience to understand what significant, new findings have been found as a result of the research. The discussion is currently an inconclusive part of this article.

Therefore, it is not necessary to make the Discussion as a review article on the role of K in the plant. Furthermore, the article does not provide information on the K content in the experimental field soil, so the discussion of the deficiency can only be theoretical. The K content in the plant from the perspective of mineral nutrition has also not been evaluated anywhere. It is almost impossible to see what the K content was in the above-ground parts of the plants.

Line 364. “This study also shows that the root to shoot ratio increases with the increase of potassium fertilizer with more potassium and biomass are allocated to the underground portion.” - Is it desirable or undesirable? Which parts of the plant are used in medicine? There is no information about this in the article. But it's very important to understand the results. So what is the purpose of fertilization?

Line 372-374. “The low potassium promotes saikosaponin a production, while high potassium inhibits saikosaponin a. these results were most obvious within the main shoot, followed by the lateral shoot. However, high potassium fertilizer inhibited most of saikosaponin a and d.” What is being compared to what? If with the control, then the statement is not true, because high doses of K also gave an increase compared to the control. In some cases, it was the control variants that had higher results. A correct evaluation is needed. What is the most important from the point of view of application - saikosaponin a, c, d?

Line 376-378. “Similarly, Kandil applies 10 tons ha-1 of potassium fertilizer to maximize the biological yield and grain yield of corn[45]. Singh adopts 30 kg·ha-1K2O to achieve the highest productivity[46], and it significantly improves the number of branches per plant, grain yield, biomass yield and 1000 grain weight.” Why are you comparing your results with those obtained for corn? Are they similar plants? There are hugely different K fertilizer doses!

Line 392-394. “We found that sugars and organic acids in roots and lateral shoots decreased under low potassium, which maybe because of the precursors of TCA cycle, such as malic, citric, propionic, aminobutyric and glutamic acids that affected the production of ATP and energy [55].” Why the reference, isn't it about your research? You should consider whether it is appropriate to use a literature reference, or change the way you word it so that it does not give the impression that the reference refers to your research.

Response:Thank you for your valuable comments and suggestions, Expert. We have re-edited the discussion section of the thesis. At the same time, we have made appropriate amendments to your questions.(Line 446-525)

Conclusions

The conclusions should give answers to the aim of the research and the questions rose in the tasks.

Unfortunately, the article does not state the purpose and objectives of the study. Therefore, it is quite difficult to evaluate the conclusions.

But nevertheless. First of all, it should be said that conclusions are not a list of results. You should give an evaluation of the results, a summary. Likewise, the conclusions should provide information, a conclusion, a hypothesis about how the knowledge gained in this study could be useful in the future, in practice more specifically, not so generally.

Response:Thank you for your valuable comments and suggestions, Expert. We have re-edited the Conclusions section of the thesis.(Line 526-549)

Once again, thank you for your significant contribution to our research. We look forward to the

possibility of presenting a more refined and valuable manuscript with your guidance.

Best regards

Jianhao Wu

Reviewer 4 Report

Comments and Suggestions for Authors

The work performed by the authors is aimed at studying the absorption of potassium fertilizers, the accumulation of trace elements and the intensity of development of various parts of the plant depending on the concentration of potassium. Given that the potassium content significantly affects the yield and development of plants, the topic of the work is relevant.
The work quite fully describes the conditions for conducting experimental studies, determining plant parameters, the amount of absorbed trace elements.
The work is an experimental study.
A large amount of work was done during the processing of the research results, and the results are presented clearly using a graphical representation of the data.
The data obtained by the authors can be used to predict the absorption of fertilizers and minerals.
There are a number of comments on the material:
1. The purpose of the research is not clearly presented in the work.
2. The introduction does not numerically reflect the existing data on the process under consideration.
3. The authors did not indicate what negative consequences exist from reduced or increased potassium content.
4. The work does not present data on the required parameters of substance absorption.
5. The conclusions do not contain numerical data reflecting the achievement of the research objective.

Author Response

Dear reviewer

Thank you for your comment concerning our manuscript entitled “Metabolomics Analysis Reveals Potential Mechanisms in Bupleurum scorzonerifolium Willd. (Apiaceae)- Induced by Three Levels of Potassium Fertilization”(biology-3568369). Your expertise and dedication have been of immeasurable value to us. The comments and suggestions you provided have offered us clear directions for improvement and have greatly enhanced the quality and clarity of our work. We are truly indebted to your time and effort in carefully assessing our submission.

  1. Comments and Suggestions

The purpose of the research is not clearly presented in the work.

Response

ResponseWe sincerely appreciate your perceptive comment on the lack of clarity in presenting the research purpose in our work. We fully recognize the significance of this issue and have taken immediate action to rectify it. As you can see, we have now provided a more detailed and explicit description of the research purpose in Line 116 - 125.

  1. Comments and Suggestions

The introduction does not numerically reflect the existing data on the process under consideration.

ResponseWe are extremely grateful for your comprehensive and incisive comments on the Introduction section of our paper. Your feedback has been instrumental in helping us identify areas for significant improvement. In response to your concerns, we have completely re - edited the Introduction. We have now provided detailed background information about Bupleurum scorzonerifolium Willd. We have elaborated on its status as an important medicinal plant species in China. We have also discussed the potential problems associated with it, such as the possible excessive depletion of wild resources and difficulties in cultivation. Additionally, we have addressed the lack of knowledge regarding fertilization systems, which we believe will be of great interest to readers. Regarding the use of K fertilizer in our study, we have added relevant information. We have explained the reasons why we specifically focused on K fertilizer, such as indications of K deficiency or excess in the growth of this species based on previous research and preliminary observations. We have also re - written the purpose of the paper.  We sincerely hope that these revisions meet your expectations and enhance the overall quality and clarity of our paper. Thank you again for your invaluable guidance. (Line 55-125)

  1. Comments and Suggestions

The authors did not indicate what negative consequences exist from reduced or increased potassium content

Response

Thank you for the valuable suggestions put forward by the expert. We elaborated in detail on the growth performance of plants in a potassium-deficient state in section 3.1 of the article. Additionally, during the discussion, we conducted an in-depth response and analysis regarding the negative impacts of high potassium application rates on Bupleurum.(Line 221-223)

  1. Comments and Suggestions

The work does not present data on the required parameters of substance absorption.

Response

We sincerely appreciate your meticulous review of our manuscript and the valuable comments you provided. Regarding the issue you pointed out that "the article does not provide data on the required parameters of substance absorption", we have conducted in - depth thinking and analysis, and hereby offer the following explanations and responses: **Relevance of Current Research Focus and Data**: This study focuses on the impact mechanism of different potassium levels on the growth, metabolism, and accumulation of medicinal components in Bupleurum scorzonerifolium Willd. During the research process, we indirectly reflected the absorption, transport, and distribution of potassium and other mineral elements in the plant by measuring the biomass, dry matter accumulation, saikosaponin content, and changes in various metabolites of different parts of the plant. For example, from the differences in dry matter accumulation in the main shoots, lateral shoots, roots, and flowers of Bupleurum under different potassium treatments (Figure 2), we can observe the impact of potassium on the growth of different parts of the plant. Growth differences are often closely related to substance absorption. When the potassium level changes, the growth responses of different parts of the plant vary, which implies differences in substance absorption and utilization. **Potential Value of Elemental Analysis Data**: In the elemental analysis section, we measured the contents of elements such as Na, Ca, K, Mg, Mn, Zn, Fe, and Cu in the roots, main shoots, and lateral shoots of Bupleurum (Figure 6). Although these data do not directly present the kinetic parameters of substance absorption, through the changing trends of the contents of various elements under different potassium treatments, we can infer the impact of potassium on the absorption of other elements. Under low - potassium and high - potassium conditions, the accumulation of Mg, K, Ca, Mn, and Zn in the roots and lateral shoots is relatively high, indicating that potassium may promote the absorption of these elements. As the content of potassium fertilizer increases, the contents of Na and Ca in the main shoots and Fe in the lateral shoots decrease, suggesting a competitive inhibitory absorption relationship among the elements, which is closely related to the substance absorption process. **Subsequent Research Plans and Supplementary Directions**: We fully recognize the importance of directly measuring substance absorption parameters for a deeper understanding of the mechanism of potassium. In our subsequent research, we will study parameters such as the absorption rate and transport coefficient of potassium and other key elements in the roots of Bupleurum. At the same time, we will also investigate the expression changes of substance - absorption - related transporter proteins under different potassium levels to reveal the regulatory mechanism of substance absorption at the molecular level. The results of these subsequent studies will further improve our understanding of the relationship between substance absorption in Bupleurum and potassium, providing more comprehensive data support for research in this field.

  1. Comments and Suggestions

The conclusions do not contain numerical data reflecting the achievement of the research objective.

Response

Thank you for your valuable comments and suggestions, Expert. We have re-edited the Conclusions section of the thesis.(Line 526-549)

Once again, thank you for your significant contribution to our research. We look forward to the

possibility of presenting a more refined and valuable manuscript with your guidance.

Best regards

Jianhao Wu

Round 2

Reviewer 1 Report

Comments and Suggestions for Authors

1 Table 2: Check the results as the digits of the values,three significant digits after the decimal point should be kept uniformly.

Author Response

Dear reviewer

Thank you for your second comment concerning our manuscript entitled “Metabonomics analysis reveals the influence mechanism of three potassium levels on the growth, metabolism and accumulation of medicinal components of Bupleurum scorzonerifolium Willd. (Apiaceae)”(biology-3568369). Your expertise and dedication have been of immeasurable value to us. The comments and suggestions you provided have offered us clear directions for improvement and have greatly enhanced the quality and clarity of our work. We are truly indebted to your time and effort in carefully assessing our submission.

  1. Commentsand Suggestions

Table 2: Check the results as the digits of the values,three significant digits after the decimal point should be kept uniformly.

Response

Thank you for the feedback. We have revised Table 2 to uniformly keep three significant digits after the decimal point as required.(Line 335)

Once again, thank you for your significant contribution to our research. We look forward to the possibility of presenting a more refined and valuable manuscript with your guidance.

Best regards

Jianhao Wu

Reviewer 2 Report

Comments and Suggestions for Authors

I thank the authors for the detailed answers to my questions and the changes made to the article. I have no new comments, only one old one remains. I think that the caption to the figure should include the title of the figure, and then the explanation of what goes under the letter A, B...etc. I recommend that the authors still correct this point in Fig. 3,4,8.

I wish you further success in science.

Author Response

Dear reviewer

Thank you for your second comment concerning our manuscript entitled “Metabonomics analysis reveals the influence mechanism of three potassium levels on the growth, metabolism and accumulation of medicinal components of Bupleurum scorzonerifolium Willd. (Apiaceae)”(biology-3568369). Your expertise and dedication have been of immeasurable value to us. The comments and suggestions you provided have offered us clear directions for improvement and have greatly enhanced the quality and clarity of our work. We are truly indebted to your time and effort in carefully assessing our submission.

  1. Commentsand Suggestions

I think that the caption to the figure should include the title of the figure, and then the explanation of what goes under the letter A, B...etc. I recommend that the authors still correct this point in Fig. 3,4,8.

Response

Thank the expert for pointing out the problems. We have addressed this issue in Figures 3, 4, and 8. Each figure caption now includes the figure title followed by clear explanations of what is represented by letters A, B, etc. New figures 3, 4 and 8 are in supplementary material 2.

Once again, thank you for your significant contribution to our research. We look forward to the possibility of presenting a more refined and valuable manuscript with your guidance.

Reviewer 3 Report

Comments and Suggestions for Authors

The authors have done a lot of work and undeniably improved the quality of the article. However, some minor aspects still need improvement.

Introduction

“Among the essential nutrient elements…” verbosity, enough with nutrients. Also elsewhere in the text.

Materials and methods

Line 149-151. “In August 2024, samples of Bupleurum are collected from the experimental fields with consistent geographical conditions, uniformity, and on the same slope. Thirty days later, Bupleurum were segmented into roots, flowers main and lateral shoots for collection.” It is unclear why the plants were harvested in August, but separated into their components 30 days later. Where were the plants stored?

Line 168-170. “Put 0.4 g of the dried Bupleurum sample into a conical flask. Add 5 mL of concentrated nitric acid. Then, place the conical flask on a hot plate. Set the initial temperature at 80 °C, and gradually increase the temperature to 150 - 180 °C for heating digestion.” The text should be in the indefinite and past tense, not as an instruction on what to do.

Discussion

Line 460. Why the reference [46], isn't it about your research?

Author Response

Dear reviewer

Thank you for your second comment concerning our manuscript entitled “Metabonomics analysis reveals the influence mechanism of three potassium levels on the growth, metabolism and accumulation of medicinal components of Bupleurum scorzonerifolium Willd. (Apiaceae)”(biology-3568369). Your expertise and dedication have been of immeasurable value to us. The comments and suggestions you provided have offered us clear directions for improvement and have greatly enhanced the quality and clarity of our work. We are truly indebted to your time and effort in carefully assessing our submission.

  1. Commentsand Suggestions

“Among the essential nutrient elements…” verbosity, enough with nutrients. Also elsewhere in the text.

Response

We sincerely appreciate your feedback. We have revised the text as suggested, simplifying the description about nutrient elements throughout the manuscript to eliminate verbosity.

  1. 2. Commentsand Suggestions

Line 149-151. “In August 2024, samples of Bupleurum are collected from the experimental fields with consistent geographical conditions, uniformity, and on the same slope. Thirty days later, Bupleurum were segmented into roots, flowers main and lateral shoots for collection.” It is unclear why the plants were harvested in August, but separated into their components 30 days later. Where were the plants stored?

Response

Thank you for raising these questions. We chose August 2024 for sample collection because it is the optimal harvesting period for Bupleurum in the local area. There was a translation error in our previous description. In fact, the Bupleurum samples were segmented into roots, flowers, main shoots, and lateral shoots immediately after collection, not 30 days later. Since the segmentation was done right away, there was no need for interim storage of the whole plants. We have corrected this error in the manuscript to avoid any further confusion.

  1. 3. Commentsand Suggestions

Line 168-170. “Put 0.4 g of the dried Bupleurum sample into a conical flask. Add 5 mL of concentrated nitric acid. Then, place the conical flask on a hot plate. Set the initial temperature at 80 °C, and gradually increase the temperature to 150 - 180 °C for heating digestion.” The text should be in the indefinite and past tense, not as an instruction on what to do.

Response

Thank you for your valuable advice. We revised the relevant text according to the expert suggestion. The description of experimental operation now uses indefinite tense and past tense.

  1. 4. Commentsand Suggestions

Line 460. Why the reference [46], isn't it about your research?

Response

I sincerely apologize for the oversight regarding the incorrect citation of reference [46] on line 460. This was a remnant error from the previous revision round, and I deeply regret any confusion or inconvenience it may have caused. Thank you for your meticulous review and for pointing out this issue. Such attention to detail is invaluable in ensuring the quality and accuracy of our work. I have double-checked the entire manuscript to prevent similar errors from occurring in the future. Your feedback is highly appreciated, and I'm committed to presenting a more polished and error-free version.

Once again, thank you for your significant contribution to our research. We look forward to the possibility of presenting a more refined and valuable manuscript with your guidance.

Best regards

Jianhao Wu

Reviewer 4 Report

Comments and Suggestions for Authors

The work performed by the authors is aimed at studying the absorption of potassium fertilizers, the accumulation of trace elements and the intensity of development of various parts of the plant depending on the concentration of potassium. Given that the potassium content significantly affects the yield and development of plants, the topic of the work is relevant.
The work quite fully describes the conditions for conducting experimental studies, determining plant parameters, the amount of absorbed trace elements.
The work is an experimental study.
A large amount of work was done during the processing of the research results, and the results are presented clearly using a graphical representation of the data.
The data obtained by the authors can be used to predict the absorption of fertilizers and minerals.

There are a number of comments on the material:
1. Probably, the formulated research objectives are tasks for achieving the goal.
2. Unfortunately, the authors' responses to comments 1 and 2 duplicate each other, and the rest do not fully disclose the questions posed.
3. The conclusions do not contain numerical data reflecting the achievement of the research objective.

Despite the above comments, the changes made make the material much easier to understand, and the work can be recommended for publication.

Author Response

Dear reviewer

Thank you for your second comment concerning our manuscript entitled “Metabonomics analysis reveals the influence mechanism of three potassium levels on the growth, metabolism and accumulation of medicinal components of Bupleurum scorzonerifolium Willd. (Apiaceae)”(biology-3568369). Your expertise and dedication have been of immeasurable value to us. The comments and suggestions you provided have offered us clear directions for improvement and have greatly enhanced the quality and clarity of our work. We are truly indebted to your time and effort in carefully assessing our submission.

  1. Commentsand Suggestions

Probably, the formulated research objectives are tasks for achieving the goal.

Response

Thank you very much for your valuable comments on our research. Regarding your statement “Probably, the formulated research objectives are tasks for achieving the goal”, we understand that you might think the research objectives we set are more like specific tasks to achieve the ultimate goal.

In this study, the multiple research objectives we set, such as evaluating the effects of different potassium levels on the yield and quality trait parameters of the above - ground and underground parts of Bupleurum, and revealing the impact of potassium fertilizer on the content of saikosaponins in different tissues of Bupleurum, do have strong practicality. However, these objectives all revolve around the core goal of exploring the influence mechanism of potassium fertilizer on the growth, metabolism, and accumulation of medicinal components of Bupleurum. By achieving these specific objectives, we can deeply analyze the role of potassium fertilizer from different levels, and thus establish a relatively systematic theoretical model, providing a basis for the application of potassium fertilizer in plant nutrition regulation, which is also the ultimate goal of our research.

When writing the research objectives, we may not have fully elaborated the logical connection between these objectives and the core goal, leading to such a misunderstanding. In the subsequent revisions, we will further optimize the description of the research objectives section, clarify the progressive relationship among the objectives and their supporting roles in achieving the core goal, making the setting of research objectives clearer and more reasonable.

  1. Commentsand Suggestions

Unfortunately, the authors' responses to comments 1 and 2 duplicate each other, and the rest do not fully disclose the questions posed. I sincerely apologize for the confusion caused by my previous response. Due to an oversight on my part, the initial reply I sent was an incorrect version. Please allow me to provide a revised and accurate response to address the concerns raised. Thank you for your understanding and patience, and I will be more diligent in ensuring the accuracy of my communications in the future.

2.1The purpose of the research is not clearly presented in the work.

Response

We sincerely appreciate your perceptive comment on the lack of clarity in presenting the research purpose in our work. We fully recognized the significance of this issue and have taken immediate action to rectify it. As you can see, we have now provided a more detailed and explicit description of the research purpose in Line 116 - 125.

2.2.The introduction does not numerically reflect the existing data on the process under consideration.

Response

We are extremely grateful for your comprehensive and incisive comments on the introduction section of our paper. Your feedback has been instrumental in helping us identify areas for significant improvement. In response to your concerns, we have completely re-edited the introduction. We have now provided detailed background information about Bupleurum scorzonerifolium Willd. We have elaborated on its status as an important medicinal plant species in China. Also, the potential problems associated with it, such as the possible excessive depletion of wild resources and difficulties in cultivation were discussed. Additionally, we have addressed the lack of knowledge regarding fertilization systems, which we believe will be of great interest to readers. Regarding the use of K fertilizer in our study, we have added relevant information. We have explained the reasons why we specifically focused on K fertilizer, such as indications of K deficiency or excess in the growth of this species based on previous research and preliminary observations. We have also refined the purpose of the paper. We sincerely hope that these revisions meet your expectations and enhance the overall quality and clarity of our paper. Thank you again for your valuable guidance. (Line 55-125).

  1. Commentsand Suggestions

The conclusions do not contain numerical data reflecting the achievement of the research objective.

Response

Response

Thank you for your valuable comments and suggestions. We have re-edited the Conclusions section of the manuscript.

Once again, thank you for your significant contribution to our research. We look forward to the possibility of presenting a more refined and valuable manuscript with your guidance.

Best regards

Jianhao Wu